# CircCDC42-encoded CDC42-165aa regulates macrophage pyroptosis in *Klebsiella pneumoniae* infection through Pyrin inflammasome activation

Nana Xu[1,2,6], Jiebang Jiang[1,6], Fei Jiang[1,3], Guokai Dong[1,2], Li Meng[1,2], Meng Wang[1,2], Jing Chen[1], Cong Li[4], Yongping Shi [1] ✉, Sisi He [5] ✉ & Rongpeng Li [1] ✉

The circular RNA (circRNA) family is a group of endogenous non-coding RNAs (ncRNAs) that have critical functions in multiple physiological and pathological processes, including inflammation, cancer, and cardiovascular diseases. However, their roles in regulating innate immune responses remain unclear. Here, we define Cell division cycle 42 (CDC42)−165aa, a protein encoded by circRNA circCDC42, which is overexpressed in *Klebsiella pneumoniae* (KP)-infected alveolar macrophages. High levels of CDC42-165aa induces the hyperactivation of Pyrin inflammasomes and aggravates alveolar macrophage pyroptosis, while the inhibition of CDC42-165aa reduces lung injury in mice after KP infection by inhibiting Pyrin inflammasome-mediated pyroptosis. Overall, these results demonstrate that CDC42-165aa stimulates Pyrin inflammasome by inhibiting CDC42 GTPase activation and provides a potential clinical target for pathogenic bacterial infection in clinical practice.

Circular RNA (circRNA) is an endogenous RNA molecule with a covalently closed structure[1,2]. In recent years, studies have increasingly suggested that circRNAs play an important role in physiological and pathological processes and are involved in the development and progression of various diseases[3,4]. In contrast to regular linear RNA, circRNAs have a closed loop shape, which confers RNA exonuclease and degradation resistance and allows for stable expression[5], making them potential targets for disease screening and therapeutic agents. To date, diverse functions of circRNAs have been elucidated, including microRNA (miRNA) sponge, regulation of translation, and protein binding[6]. Numerous studies have identified small open reading frames (sORFs) in circRNAs, which enable them to encode proteins under physiological and environmental stress conditions such as hypoxia,

heat shock, or viral infection, and therefore play important roles in regulating cancer and cardiovascular diseases[7]. However, the roles of circRNA in the regulation of bacterial or viral infection-related innate immune responses have not been well understood.

*Klebsiella pneumoniae* (KP) is the most common Gram-negative opportunistic pathogen that often infects people with impaired defense mechanisms or weakened immunity and is considered a leading pathogenic cause of community-acquired pneumonia and nosocomial infections[8]. In the past decades, increasing numbers of highly virulent KP (HvKP) strains have been identified in hospitals, which pose a serious threat to public health and the effectiveness of clinical therapeutics. This trend highlights an urgent need to explore new screening and therapeutic targets for KP infection. KP has

[1]Jiangsu Province Engineering Research Center of Cardiovascular Drugs Targeting Endothelial Cells, School of Life Sciences, Jiangsu Normal University, Xuzhou, China. [2]Laboratory of Morphology, Xuzhou Medical University, Xuzhou, China. [3]Department of Laboratory Medicine, the Affiliated Hospital of Xuzhou Medical University, Xuzhou, China. [4]Xuzhou Key Laboratory of Emergency Medicine, the Affiliated Hospital of Xuzhou Medical University, Xuzhou, China. [5]Department of Oncology, The Second Affiliated Hospital of Zunyi Medical University, Zunyi, China. [6]These authors contributed equally: Nana Xu, Jiebang Jiang. ✉e-mail: shiyongp@sina.com; sisihe1219@163.com; lirongpeng@jsnu.edu.cn

traditionally been considered an extracellular pathogen that is phago-cytosed by alveolar macrophages (AM). However, HvKP has a thick capsule and exhibits resistance to complement and neutrophil-mediated phagocytosis[9]. In addition, HvKP can survive and escape from macrophages by limiting the fusion of *Klebsiella*-containing vacuoles (KCVs) and lysosomes, thereby escaping immune clearance[10,11]. Investigating the biological process and molecular mechanism of the interaction between KP and host macrophages may identify a specific target for the precise regulation of the function of AMs, which has important theoretical implications in boosting the host immune system response to KP lung infection. Pyroptosis is a type of highly pro-inflammatory programmed cell death induced by inflammasomes. Although recent studies have highlighted the potential role of pyr-optosis in the pathogenesis of KP infection[12,13], the crosstalk between AMs and pyroptosis during KP infection and the regulatory mechanisms are largely unknown.

Cell division cycle 42 (CDC42) is a small GTPase belonging to the Ras homologous Rho family. It regulates various cellular functions after being activated by guanine nucleotide exchange factors (GEFs), such as proliferation, migration, enzyme activity, and cell polarity[14]. The dedicator of cytokinesis (DOCK) family proteins serve as important upstream GEFs that activate the small Rho GTPases CDC42 and Ras-related C3 botulinum toxin substrate 1 (RAC1), among which only DOCK8 is a CDC42-specific GEF that plays an important role in functions of diverse immune cell[15–17]. Recent studies have also shown that CDC42 is crucial for the action of Pyrin inflammasomes to counter invading pathogens, such as *Clostridium difficile*, *Yersinia pestis*, and *Burkholderia cenocepacia*, thereby inducing macrophage pyroptosis[18–20]. However, the role of the *CDC42* gene and its splicing products in KP infection has not been elucidated.

In this work, we present a translatable circRNA (circCDC42), which is generated by back-splicing of *CDC42* and overexpressed in KP-infected AMs. We demonstrate that circCDC42 encodes a protein known as CDC42-165aa that contains 165 amino acids. We find that CDC42-165aa is an important molecular decoy that inhibits CDC42 activity by competitively binding to DOCK8, thereby inducing Pyrin-mediated pyroptosis. Our data indicate that circCDC42 is a functional CDC42 mutation that encodes CDC42-165aa, a positive regulator of the pyrin-mediated pyroptosis pathway.

## Results

### CircCDC42 expression is increased in murine AMs during KP infection

RNA-seq technology was utilized to analyze differentially expressed circRNAs (DEcRs) to investigate the impact of host circRNAs on the respiratory innate immune response in AMs during KP infection. Seventy-five upregulated and 50 downregulated circRNAs were identified in KP-infected MH-S cells compared with uninfected wild-type (WT) cells (Supplementary Fig. 1a). Meanwhile, the intersection of DEcRs was searched using IRESfinder, ORFfinder, and circBase datasets, and three circBase annotated circRNAs (mmu_circ_0004602, mmu_circ_0007693, and mmu_circ_0011319) were identified, which exhibited significant upregulation and displayed potential translation function (Fig. 1a). The elevated messenger RNA (mRNA) levels of the three circRNAs in MH-S cells were further confirmed through real-time quantitative PCR (RT-qPCR). Notably, mmu_circ_0011319 displayed the most pronounced differential expression in the context of KP infection. (Fig. 1b and Supplementary Fig. 1b).

Mmu_circ_0011319 was generated by back-splicing of exons 2–5 of *CDC42* pre-mRNA with 586-nucleotide (thus termed as circCDC42) (Fig. 1c). Divergent primers were utilized to amplify the back-splice junction site of circCDC42, which was subsequently confirmed through Sanger sequencing (Fig. 1d). Agarose gel electrophoresis of the PCR product revealed that circCDC42 was only amplifiable using divergent primers in complementary DNA (cDNA), with no

amplification observed in genomic DNA (gDNA) samples (Fig. 1e), demonstrating that its closed loop structure was generated from back-splicing. Actinomycin D (Act-D) is an RNA synthesis inhibitor that can be used to analyze RNA half-life[21]. Hence, we assessed the half-life of circCDC42 and CDC42 mRNA in Act-D-treated MH-S cells and found that circCDC42 had a longer half-life than CDC42 mRNA (Fig. 1f). In addition, compared with CDC42 mRNA, circCDC42 was resistant to RNase R exonuclease digestion, indicating higher stability of circCDC42 (Fig. 1g). The subcellular localization of circCDC42 was further investigated using fluorescence in situ hybridization (FISH) and RT-qPCR. CircCDC42 was more prevalent in the cytoplasm than in the nucleus (Fig. 1h, i). CircCDC42 was almost not expressed in pulmonary epithelial cells because PCR products of the circCDC42 transcript were only obtained from AMs MH-S but not epithelial MLE-12 cells (Supplementary Fig. 1c).

Next, the expression pattern of circCDC42 was compared in different tissues and cell types to determine whether the increase in circCDC42 was tissue- or cell-type-specific. RT-qPCR data showed that the local level of circCDC42 was highest in mouse lungs and increased most significantly after KP infection compared with that in brain, liver, spleen, and kidney tissues (Supplementary Fig. 1d, e). RT-qPCR and FISH results revealed that circCDC42 was mainly expressed in AMs rather than in neutrophils, T cells, or NK cells, especially after KP infection (Fig. 1j, k, and Supplementary Fig. 1f). Together, these data suggested that circCDC42 plays a dominant role in AMs of the lung tissues. Various bacterial strains and species were examined to determine whether circCDC42 was only involved in KP infection. RT-qPCR results showed that circCDC42 was increased in MH-S cells respectively infected with *Escherichia coli* (*E. coli*), *Pseudomonas aeruginosa* (PAO1), and *Staphylococcus aureus* (Supplementary Fig. 1g), suggesting that cirCDC42 may play a role in a wide range of Gram-negative bacteria.

### CircCDC42 encodes a 165-amino-acid protein, CDC42-165aa

The protein-coding potential calculator (CPC)[22] and ORF finder analysis[23] revealed that circCDC42 contains an ORF with ATG initiation codons and an internal ribosome entry site (IRES) at 309–415 nt (Fig. 2a and Supplementary Fig. 2a), suggesting that circCDC42 has a potential capacity to encode a 165aa protein. To identify the activity of the predicted IRES in circCDC42, we constructed a WT IRES or mutated IRES sequence into a dual luciferase reporter gene vector. Contrary to the mutated IRES, the full-length circCDC42 IRES induced the highest level of Luc/Rluc activity (Fig. 2b). These results confirmed that the circCDC42 IRES could drive 5′-cap independent translation. Driven by this IRES, a cross-junction and predicted ORF potentially encoded a putative 165aa protein with four unique sequences, which was named CDC42-165aa in the present study. Next, we used an antibody against the middle part (39-89aa) of CDC42, which simultaneously identified CDC42 and CDC42-165aa, according to different molecular weights in the immunoblotting assay (Fig. 2c). The levels of CDC42 and CDC42-165aa in MH-S cells and primary AMs isolated from bronchoalveolar lavage fluid (BALF) of mice were detected by immunoblotting assay. The results showed that CDC42-165aa was highly expressed in both MH-S cells and AMs after KP infection (Fig. 2d and Supplementary Fig. 2b). Furthermore, to confirm whether CDC42-165aa was encoded by circCDC42, MH-S cells were transfected with a circCDC42 over-expression plasmid − a linearized CDC42-165aa overexpression plas-mid with FLAG tag (CDC42-165aa-Flag) − and a circCDC42 ATG delete (circCDC42 ATG-del) plasmid (the initiation codon of circCDC42 was deleted). RT-qPCR results showed that circCDC42 and circCDC42 ATG-del plasmids effectively increased the levels of WT and mutated circCDC42, while CDC42-165aa plasmids did not show such changes (Fig. 2e and Supplementary Fig. 2c). Immunoblotting results showed that transfection of both circCDC42 and CDC42-165aa-Flag plasmids produced the predicted CDC42-165aa band in MH-S cells, while

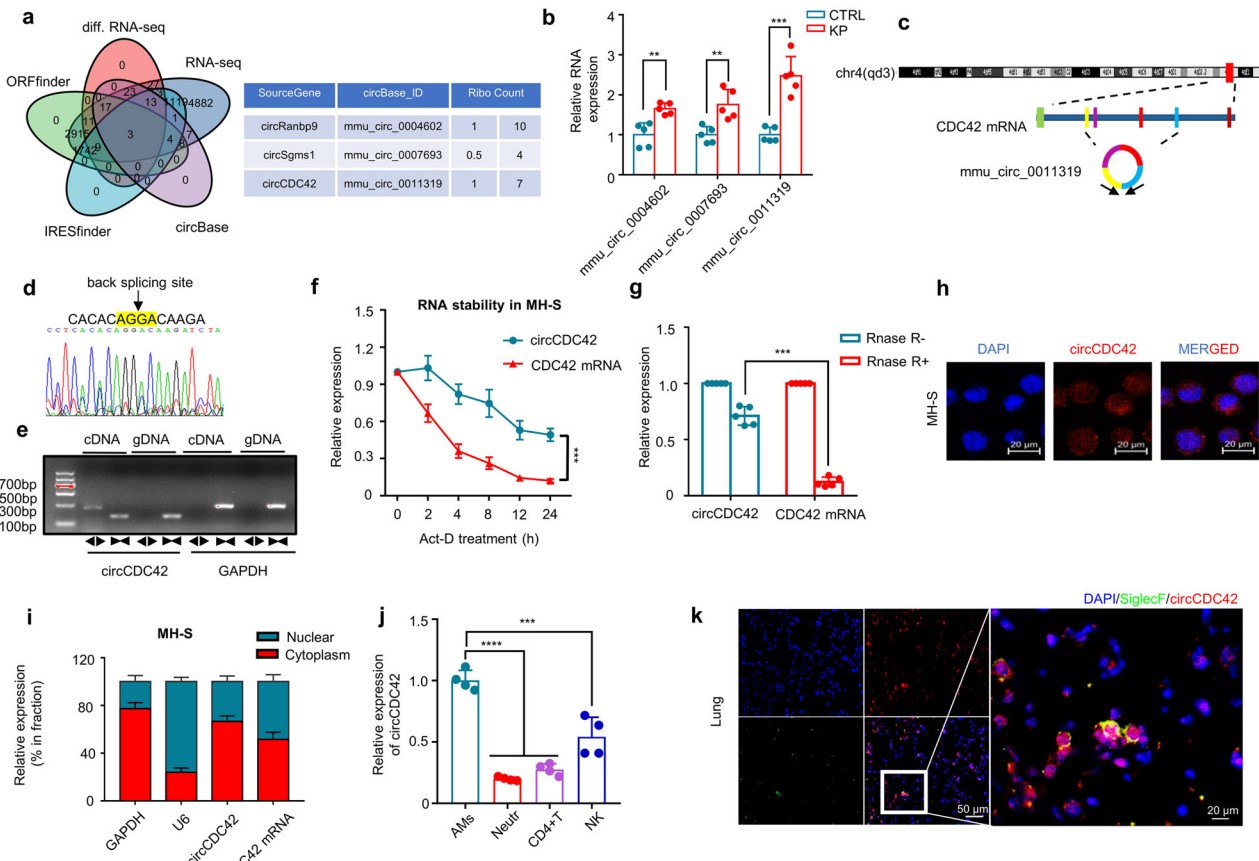

**Fig. 1 | circCDC42 is a potential coding circRNA and is overexpressed in MH-S following KP infection. a** Three pairs of MH-S cells infected with or without KP (MOI of 20:1) were prepared for RNA sequencing (RNA-seq). The intersection DEcRs of the candidate circRNAs with the IRESfinder, ORFfiner, and circBase datasets was determined to screen for translatable circRNAs. Left: Venn diagram showing the coding circRNAs and DEcRs intersecting in MH-S cells. Right: three circBase annotated circRNAs with coding potential and their corresponding RPFs. RPFs, ribosome protected frames. **b** Validation of the expression levels of circRNAs in MH-S cells following KP infection by RT-qPCR (*n* = 5 biologically independent samples). **c** A schematic diagram showing the circCDC42 (mmu_circ_0011319) formed from exons 2-5 of *CDC42* gene. **d** Sanger sequencing results verifying head-to-tail splicing of circCDC42 in MH-S. **e** Complementary DNA (cDNA) and genomic DNA (gDNA) served as templates in the amplification of circRNAs in MH-S cells using divergent and convergent primers. A representative photograph of agarose gel electrophoresis after PCR (*n* = 3 biological replicates). **f** MH-S cells were treated with actinomycin D (2 µg/mL) and used for RNA extraction and RT-qPCR analysis of circRNAs at the indicated time points (*n* = 3 biological replicates). **g** Total RNAs extracted from MH-S cells were treated with or without RNase R for 1 h and then subjected to RNA extraction and RT-qPCR analysis (*n* = 5 biological replicates). **h** Fluorescence in situ hybridization (FISH) with junction-specific probes was conducted to determine the localization of circCDC42 (*n* = 3 biological replicates). Scale bars, 20 µm. **i** MH-S cells were subjected to nuclear and cytoplasmic separation, and circCDC42 or linear *CDC42* mRNA were detected by RT-qPCR (*n* = 3 biological replicates). **j** The expression level of circCDC42 in major immune cells, including neutrophils, lymphocytes, NK cells, and alveolar macrophages as determined by RT-qPCR (*n* = 4 biological replicates). **k** The FISH analysis of mouse lungs to determine the localization of circCDC42 in AMs. SiglcF and circCDC42 probes were used to label AMs and circCDC42, respectively (*n* = 3 biological animals). Scale bar, 20 µm. Data are presented as means ± SD. **b**, **f**, **g**, **i** Two-way ANOVA with Bonferroni's test. **j** One-way ANOVA with Tukey test. \*\**p* < 0.01; \*\*\**p* < 0.001; \*\*\*\**p* < 0.0001. *p*-values: (**b**) *p* = 0.0075 (mmu_circ_0004602), *p* = 0.0019 (mmu_circ_0007693), *p* = 7.09E-08 (mmu_circ_0011319). (**f**) *p* = 2.69E-04. (**g**) *p* = 2.28E-07. (**j**) *p* = 8.38E-06 (neutr), *p* = 2.34E-05 (CD4 + T), *p* = 0.00159 (NK). Also, see Figure S1. Source data are provided as a Source data file.

overexpression of the circCDC42 ATG-del plasmid did not show such changes (Supplementary Fig. 2d). Subsequently, the sequence was detected using mass spectrometry followed by silver staining, which was consistent with the putative CDC42-165aa sequence containing the four unique C-terminal sequences (Fig. 2f). CDC42-165aa-Flag was mainly found in the cytoplasm of MH-S cells as determined by immunofluorescence using anti-Flag antibody (Fig. 2g). In addition, the expression of CDC42-165aa was examined at different time points after KP infection to explore the transcription and expression kinetics of CDC42-165aa in KP-infected MH-S cells and AMs. It was found that the mRNA of circCDC42 and protein level of CDC42-165aa peaked at 4–8 h (Supplementary Fig. 2e, f), while the cell viability was significantly decreased (Supplementary Fig. 2g). Moreover, another CDC42 antibody against AA166-182 did not detect the CDC42-165aa band (Supplementary Fig. 2f), further confirming that CDC42-165aa lacks the C-terminus of CDC42. CDC42-165aa was highly expressed in AMs

isolated from BALF of mice after KP infection for 24 h (Supplementary Fig. 2h).

## CDC42-165aa regulates pyroptosis of alveolar macrophages

Next, small interfering RNAs (siRNAs) that specifically target the back-splice region of circCDC42 were generated to explore the biological function of CDC42-165aa following KP infection. The siRNAs were transferred into MH-S cells, and RT-qPCR results showed that siRNA-1 effectively decreased the level of circCDC42 but had no effect on the mRNA level of CDC42 (Supplementary Fig. 3a). In addition, circCDC42 overexpression (circCDC42-OE) plasmid was transfected into MH-S cells (Supplementary Fig. 3b). As shown in Fig. 3a and Supplementary Fig. 3c, overexpression of circCDC42 significantly elevated the CDC42-165aa level upon KP infection in MH-S cells, while circCDC42 silencing reduced CDC42-165aa expression. Subsequent MTT (3-[4,5-dimethylthiazol-2-yl]-2,5 diphenyl tetrazolium bromide) assay showed that circCDC42

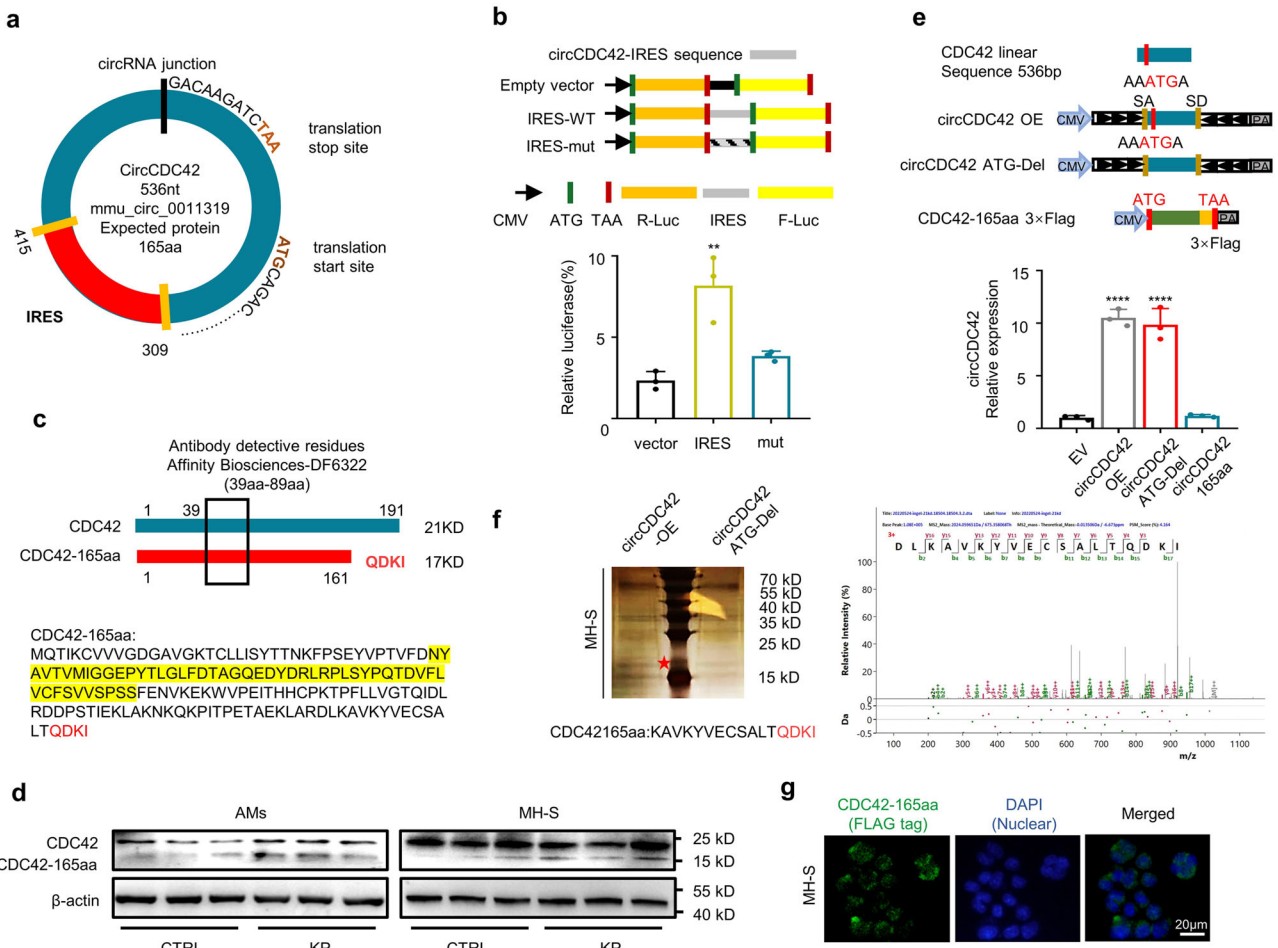

**Fig. 2 | CircCDC42 encodes a 165aa protein termed as CDC42-165aa. a** The schematic diagram shows the putative ORF and IRES in circCDC42. The splice site is shown in the black line. The unique C-terminus of CDC42-165aa was generated by an ORF spanning this splice site in circCDC42. **b** Detection of the predicted IRES activity of circCDC42. Upper panel, the wildtype or mutated IRES sequences in circCDC42 cloned between Rluc and Luc reporter genes with independent start (ATG) and stop (TAA) codons. Lower panel, the relative luciferase activity of Luc/Rluc in the above vectors (*n* = 3 biological replicates). **c** Illustration of the *CDC42* sequence and CDC42-165aa sequence. The antibody generated against the indicated sequence (in yellow) was used to recognize both proteins. **d** Both of MH-S cells and alveolar macrophages (AMs) isolated from BLAF were infected with or without KP at an MOI of 20:1 for 4 h. The expression of CDC42 and CDC42-165aa in each cell was detected by immunoblotting (*n* = 3 biological replicates). **e** Upper panel, the schematic diagram of overexpression circCDC42, the ATG-Del

circCDC42 construct, and the linearized CDC42-165aa-3×Flag construct. Lower panel, the empty vector (EV), circCDC42-OE, circCDC42-ATG-Del, and CDC42-165aa-Flag were transfected into MH-S cells for 24 h, respectively. The RT-qPCR results show the relative expression of circCDC42 in MH-S cells (*n* = 3 biological replicates). **f** Mass spectrometry analysis of the unique C-terminal protein sequence of CDC42-165aa in MH-S cells. Left panel, the total protein of MH-S cells transfected with circCDC42-OE and circCDC42 ATG-Del plasmids was separated by SDS-PAGE. Right panel, differential gel band between 17 and 25 kDa were cut and analyzed by MS. **g** Flag-tagged CDC42-165aa was transfected into MH-S cells for 24 h. Immunofluorescence staining illustrating the cellular localization of CDC42-165aa (*n* = 3 biological replicates). Scale bar, 20 μm. Data are presented as the means ± SD. **b**, **e** One-way ANOVA with Tukey test. **p < 0.01; ****p < 0.0001. See also Figure S2. Source data are provided as a Source data file.

overexpression strongly inhibited MH-S cell viability after KP infection, which was significantly enhanced after circCDC42 silencing (Fig. 3b). Pathogenic bacteria cause multiple modes of host cell death, including apoptosis, pyroptosis, and programmed necrosis. Thus, flow cytometry analysis was performed on KP-infected AMs to detect the main cell death type caused by high levels of circCDC42 expression. It was found that propidium iodide (PI)-positive cells accounted for the vast majority of KP-infected AMs, while Annexin-V + /PI- cells accounted for only a small fraction (Supplementary Fig. 3d–f). PI positive is generally considered to be the main marker of pyroptosis[24–26]. In addition, Kyoto Encyclopedia of Genes and Genomes (KEGG) data showed that CDC42-165aa displayed a notable association with *Yersinia*-type bacterial infection (a classical pyroptosis signaling pathway) (Supplementary Fig. 3g), suggesting that CDC42-165aa is mainly associated with cell pyroptosis.

To investigate the effect of CDC42-165aa on cell pyroptosis and its related inflammatory responses, diverse tests were carried out to

compare the different phenotypes in circCDC42 overexpression and knockdown MH-S cells. Light microscope observations showed that KP infection induced MH-S cell swelling and membrane rupture, which was the characteristic morphology of pyroptosis. Overexpression of circCDC42 further aggravated pyroptosis. However, circCDC42 knockdown significantly inhibited pyroptosis (Fig. 3d and Supplementary Fig. 3h). The lactate dehydrogenase (LDH) release assay showed that circCDC42 overexpression significantly aggravated KP-induced LDH release in MH-S cells, while circCDC42 silencing reduced LDH release (Fig. 3e). Immunoblotting results revealed hyper-induced pyroptosis evidenced by excessively increased Caspase-1 (CASP-1) cleavage, Gasdermin D (GSDMD) cleavage, and interleukin-1 beta (IL-1β) release in circCDC42 overexpressed cells following KP infection, while circCDC42 knockdown alleviated pyroptosis-related proteins (Fig. 3f and Supplementary Fig. 3i). Enzyme-linked immunosorbent assay (ELISA) results also showed that circCDC42 overexpression

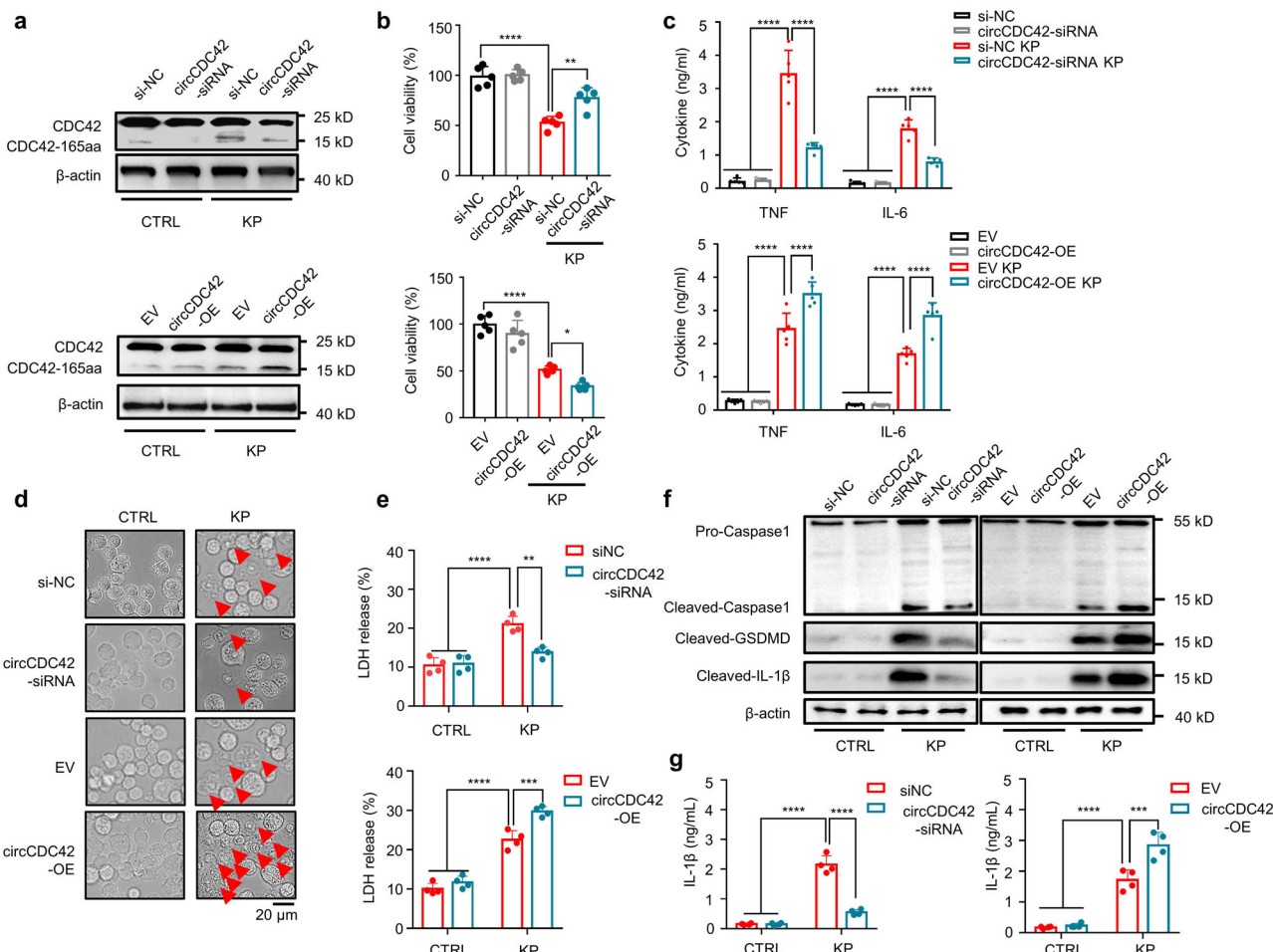

**Fig. 3 | CDC42-165aa promoted KP-induced MH-S macrophage pyroptosis MH-S cells were transfected with siNC or circCDC42 siRNA, empty vector (EV) or circCDC42 overexpreesion (OE) plasmid, respectively for 24 h. The cells were subsequently infected with KP at an MOI of 20:1 for 4 h. a** The expression of CDC42 and CDC42-165aa in the indicated MH-S was detected by immunoblotting ($n = 3$ biological replicates). **b** Results of the MTT assay showing the viability of MH-S cells ($n = 5$ biological replicates). **c** The ELISA results indicating the levels of TNF and IL-6 in the MH-S cell supernatants (n = 5 biological replicates). **d** Light microscopy analysis of the morphology of MH-S cells treated as indicated, the red arrows indicate pyroptotic cells ($n = 3$ biological replicates). Scale bar, 20 μm. **e** The release of LDH from MH-S cells was treated as indicated and presented as a percent of medium activity compared to cell lysate from cells ($n = 4$ biological replicates). **f** Immunoblotting analysis of the expression of Caspase-1, GSDMD and IL-1β in the indicated MH-S cells ($n = 3$ biological replicates). **g** The secretion of IL-1β in the

indicated MH-S cell supernatants was detected by ELISA ($n = 4$ biological replicates). Data are presented as the means ± SD. **b** One-way ANOVA with Tukey test. **c**, **e**, **g** Two-way ANOVA with Bonferroni's test. *$p < 0.05$; **$p < 0.01$; ***$p < 0.001$; ****$p < 0.0001$. *p*-values: (**b**) $p = 1.86E-06$ (siNC *vs* siNC + KP), $p = 1.26E-06$ (siNC + KP *vs* si-circCDC42 + KP), $p = 2.92E-06$ (EV *vs* EV + KP), $p = 5.23E-05$ (EV + KP *vs* OE + KP). (**c**) TNF: $p = 1.23E-10$ (siNC *vs* siNC + KP), $p = 3.15E-10$ (siNC + KP *vs* si-circCDC42 + KP), $p = 5.54E-09$ (EV *vs* EV + KP), $p = 8.96E-05$ (EV + KP *vs* OE + KP); IL-6: $p = 5.44E-10$ (siNC *vs* siNC + KP), $p = 1.50E-05$ (siNC + KP *vs* si-circCDC42 + KP), $p = 9.21E-07$ (EV *vs* EV + KP), $p = 3.08E-05$ (EV + KP *vs* OE + KP). (**e**) $p = 4.63E-06$ (siNC *vs* siNC + KP), $p = 0.0011$ (siNC + KP *vs* si-circCDC42 + KP), $p = 2.29E-07$ (EV *vs* EV + KP), $p = 0.0004$ (EV + KP *vs* OE + KP). (**g**) $p = 6.01E-10$ (siNC *vs* siNC + KP), $p = 0.0011$ (siNC + KP *vs* si-circCDC42 + KP), $p = 1.95E-05$ (EV *vs* EV + KP), $p = 0.0004$ (EV + KP *vs* OE + KP). See also Figure S3. Source data are provided as a Source data file.

promoted IL-1β secretion, which was inhibited after circCDC42 knockdown (Fig. 3g). Collectively, these results demonstrate that the KP-induced hyperactive inflammatory response was caused by pyroptosis, which was further exacerbated by circCDC42-encoded CDC42-165aa in KP-infected macrophages.

To demonstrate that CDC42-165aa and not circCDC42 induced the above functions, we transfected the circCDC42 ATG-del and CDC42-165aa-linearized plasmids into MH-S cells as the negative and positive control, respectively. After KP infection, overexpression of circCDC42 and CDC42-165aa markedly increased CDC42-165aa expression and reduced cell viability (Fig. 4a, b). However, the deletion of the circCDC42 ATG sequence reduced CDC42-165aa expression and restored cell viability to a similar level of WT MH-S cells (Fig. 4a, b, and Supplementary Fig. 4a). In addition, the deletion of the circCDC42 ATG sequence in MH-S cells did not significantly alter the cell response to KP-induced pyroptosis. (Fig. 4c–f and Supplementary Fig. 4b, c).

Overall, these data demonstrate that CDC42-165aa is critical for the regulation of KP-induced pyroptosis.

## CDC42-165aa inhibits CDC42 activity by competitively binding to DOCK8

To explore the molecular mechanism by which CDC42-165aa regulates pyroptosis, we transfected the CDC42-165aa-Flag plasmid into MH-S cells to identify proteins that bound to CDC42-165aa. Interacting proteins were pulled down in the CDC42-165aa-Flag immunoprecipitation complex and separated via SDS-PAGE (Fig. 5a). Differentially expressed proteins were identified using liquid chromatography with tandem mass spectrometry (LC-MS/MS). DOCK8 ranked highest among the identified proteins (Fig. 5b). The direct interaction between CDC42-165aa-Flag and DOCK8 was confirmed by co-immunoprecipitation (co-IP) (Fig. 5c). Subsequent immunofluorescence staining of c-Flag-transfected MH-S cells further showed the co-localization of CDC42-165aa with DOCK8

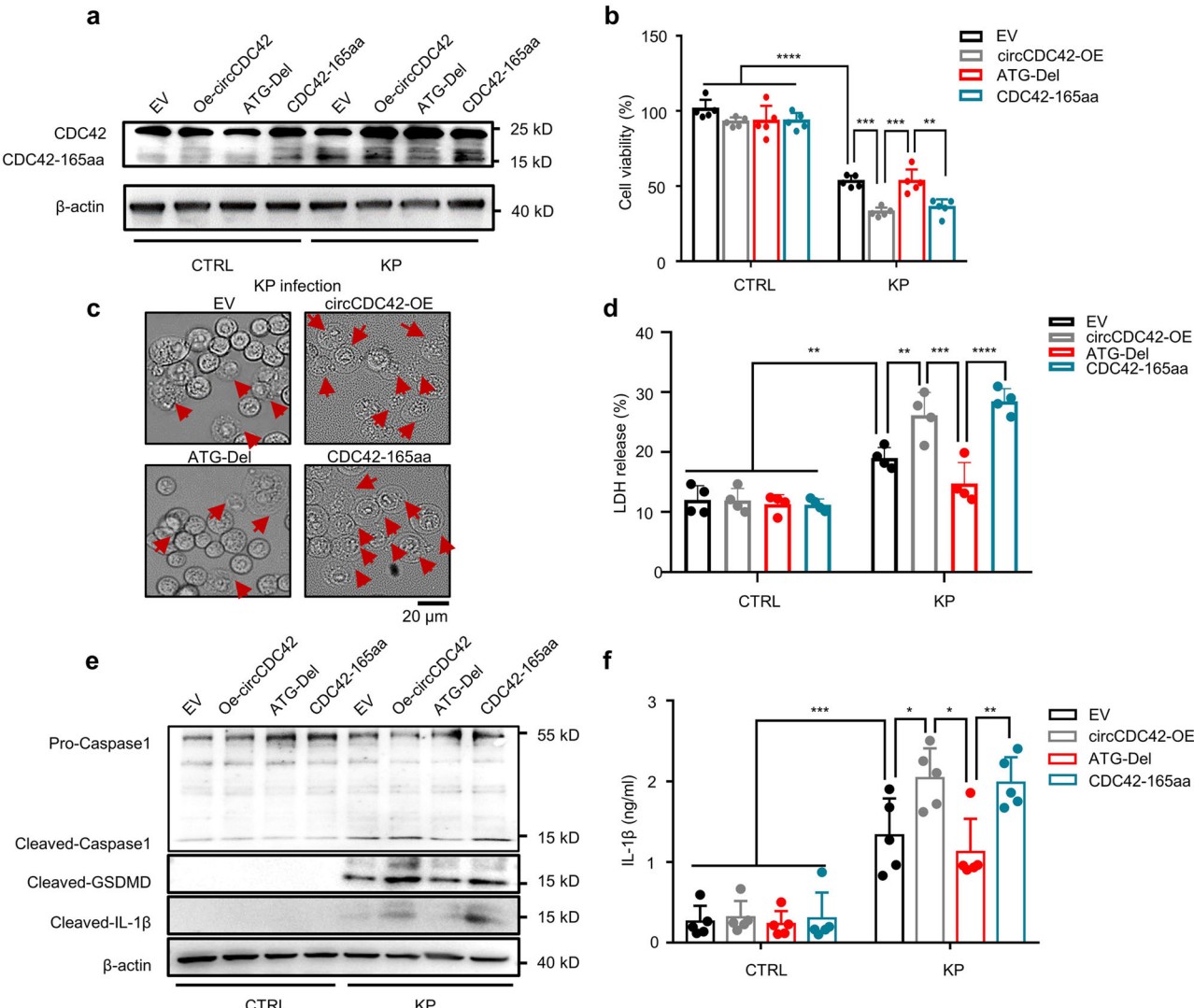

**Fig. 4 | CDC42-165aa, encoded by circCDC42, exerts the biological function.**
MH-S cells were transfected with empty vector (EV), circCDC42 overexpression
(OE) plasmid, circCDC42 ATG-Del plasmid, or CDC42-165aa plasmid, respectively
for 24 h. The cells were infected with KP at an MOI of 20:1 for 4 h. **a** Immunoblotting
results showing the expression of CDC42 and CDC42-165aa in MH-S in the indicated
groups (*n* = 3 biological replicates). **b** Measurement of the viability of the indicated
MH-S cells by MTT assay (*n* = 5 biological replicates). **c** The morphology of MH-S
cells in the indicated groups was examined using light microscopy, the red arrows
indicate pyroptotic cell (*n* = 3 biological replicates). Scale bar, 20 μm. **d** LDH release
from MH-S cells in the indicated groups, presented as percent of medium activity

compared to cell lysate from cells (*n* = 4 biological replicates). **e** Immunoblotting
analysis of the expression of Caspase-1, GSDMD, and IL-1β in MH-S cells in the
indicated groups (*n* = 3 biological replicates). **f** The secretion of IL-1β in the indi-
cated MH-S cell supernatants was detected by ELISA (*n* = 5 biological replicates).
Data are presented as means ± SD. **b**, **d**, **f** Two-way ANOVA with Bonferroni's test.
**p* < 0.05; ***p* < 0.01; ****p* < 0.001, *****p* < 0.0001. *p*-values: (**b**) *p* = 1.30E-10 (EV *vs*
EV + KP), *p* = 0.0003 (EV + KP *vs* OE + KP), *p* = 0.0003 (OE + KP *vs* ATG + KP),
*p* = 0.0029 (ATG + KP *vs* 165aa+KP). (**d**) *p* = 0.0085 (EV *vs* EV + KP), *p* = 0.0072
(EV + KP *vs* OE + KP), *p* = 0.0003 (OE + KP *vs* ATG + KP), *p* < 0.0027 (ATG + KP *vs*
165aa + KP). See also Figure S4. Source data are provided as a Source data file.

(Fig. 5d). Taken together, these results consistently demonstrated that
DOCK8 is the main CDC42-165aa-binding protein.

DOCK8 is an atypical GEF that specifically activates CDC42 by
exchanging bound guanosine diphosphate (GDP) for free guanosine
triphosphate (GTP)[16,27]. Gene ontology (GO) enrichment and Reactome
pathway analysis revealed multiple differentially expressed genes
involved in the regulation of small GTPase CDC42-related functions,
including activation of CDC42, small GTPase binding, and recruitment
and activation of neural Wiskott-Aldrich syndrome protein (N-WASP)
by the Cdc42 pathway (Supplementary Fig. 5a). Given that CDC165aa
has a similar amino acid sequence to the C-terminal 1-161 of full-length
CDC42, we proposed that CDC42-165aa may function as a decoy
molecule that competitively binds to the DOCK8, thus affecting Cdc42
activity. To test this hypothesis, we evaluated the biological activities
of CDC42 in circCDC42 overexpressed or knockdown MH-S cells,

respectively. Immunoblotting results showed that circCDC42 over-
expression dramatically inhibited CDC42 activation in MH-S cells after
KP infection, whereas circCDC42 silencing markedly enhanced the
CDC42 activity, indicating a negative regulation of CDC42 activity by
CDC42-165aa (Fig. 5e and Supplementary Fig. 5b). Furthermore,
circCDC42 overexpression substantially decreased the binding of
CDC42 to DOCK8 in MH-S cells (Fig. 5f). Conversely, circCDC42
knockdown significantly enhanced the binding of CDC42 to DOCK8
(Fig. 5g). These data further confirmed that CDC42-165aa inhibits the
activation of CDC42 by competitively interacting with DOCK8.
To confirm the importance of DOCK8 in KP-induced pyroptosis,
cell viability and the secretion of supernatant IL-1β were measured
in DOCK8 knockdown MH-S cells. The results showed that
DOCK8 knockdown significantly increased cell death and IL-1β release
(Supplementary Fig. 5c–e).

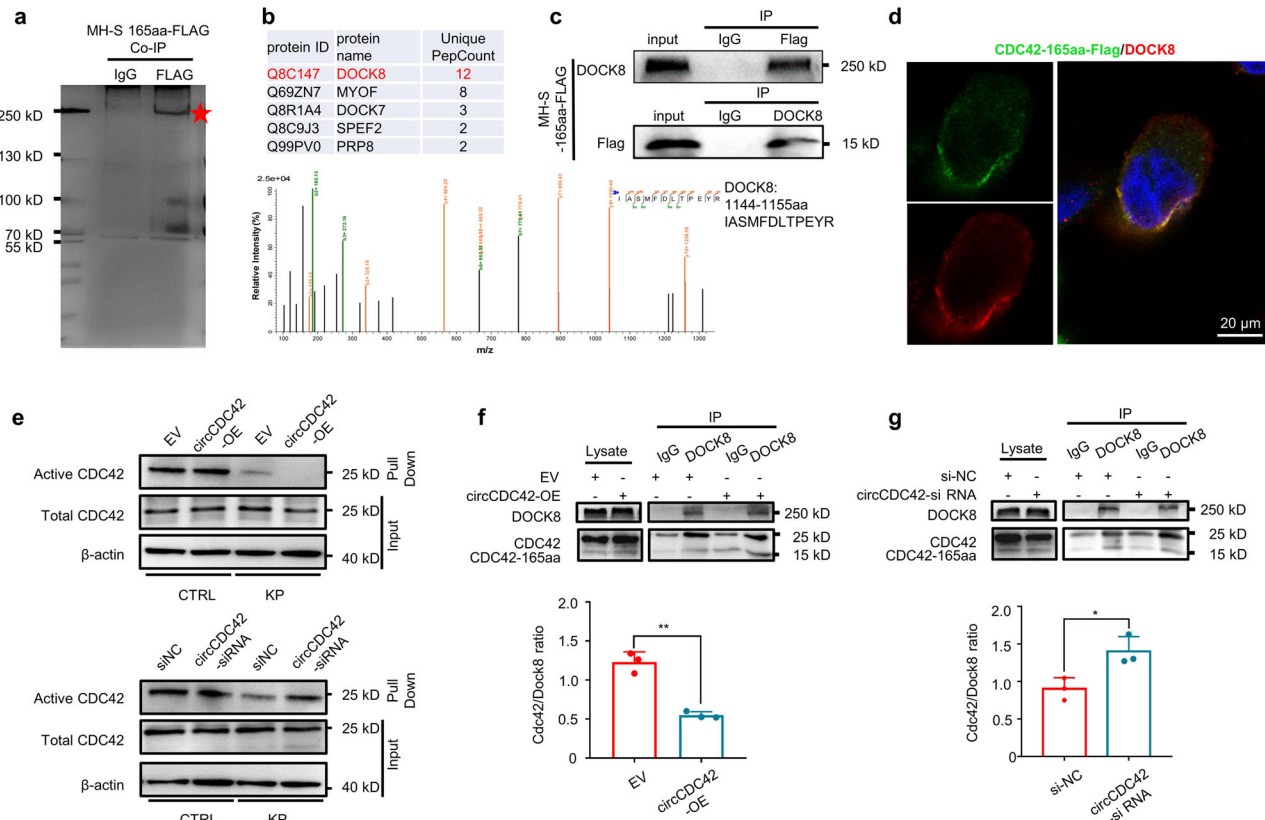

**Fig. 5 | CDC42-165aa inhibits activity of CDC42 by competitively binding to Dock8. a** MH-S cells were transfected with CDC42-165aa-Flag plasmid for 24 h, then the total protein from the indicted MH-S cells were separated via SDS-PAGE. A red arrow indicates an intense band on the gel, which was identified as CDC42-165aa binding Protein (*n* = 3 biological replicates). **b** DOCK8 peptide sequences were identified by LC/LC-MS in the CDC42-165aa protein complex. **c** Flag-tagged CDC42-165aa plasmid was transfected into MH-S cells for 24 h. CDC42-165aa binding to DOCK8 was detected by co-immunoprecipition with the anti-flag and DOCK8 antibodies, respectively (*n* = 3 biological replicates). **d** Flag-tagged CDC42-165aa plasmid was transfected into MH-S cells for 24 h. Immunofluorescence staining using anti-Flag and anti-DOCK8 antibodies was conducted to determine the co-localization of CDC42-165aa and DOCK8 (*n* = 3 biological replicates). Scale bar, 20 μm. **e** MH-S cells were transfected with siNC or circCDC42 siRNA, empty vector (EV), or circCDC42 overexpression (OE) plasmid, respectively for 24 h. The cells were infected with KP at an MOI of 20:1 for 4 h. The activated form of CDC42 and total CDC42 in MH-S cells in the indicated groups was detected by immuno-blotting assay (*n* = 3 biological replicates). **f** The expression level of CDC42-165aa bound to DOCK8 increased in MH-S cell lines transfected circCDC42-OE plasmid following KP infection (*n* = 3 biological replicates). **g** The expression level of CDC42-165aa bound to DOCK8 decreased in MH-S cell lines with knockdown of circCDC42 after KP infection (*n* = 3 biological replicates). Data are presented as means ± SD. **f**, **g** Unpaired two-tailed t-test. *$p < 0.05$; **$p < 0.01$. *p*-values: (**f**) $p = 0.0079$ (EV *vs* EV + KP), (**g**) $p = 0.0179$ (siNC *vs* si-circCDC42). See also Figure S5. Source data are provided as a Source data file.

## CDC42-165aa promotes pyroptosis in AMs by activating Pyrin inflammasome

Pyroptosis is an important form of programmed cell death accompanied by inflammasome activation and consistent inflammatory responses[28]. To determine inflammasomes associated with CDC42-165aa-induced pyroptosis, we assessed the expression of four inflammasome signature proteins in MH-S cells, including absent in melanoma 2 (AIM2), NLR family CARD domain-containing protein 4 (NLRC4), NLR family pyrin domain containing 3 (NLRP3), and pyrin. Immunoblotting results showed that CDC42-165aa overexpression increased the pyrin expression in MH-S cells after KP infection. Notably, although NLRP3 expression was increased after KP infection, CDC42-165aa overexpression did not affect its expression (Supplementary Fig. 6a). Besides, no increase in NLRC4 and AIM2 expression was observed after CDC42-165aa overexpression (Supplementary Fig. 6a). ELISA results also showed that AIM2 or NLRP3 knockdown did not affect CDC42-165aa-induced IL-1β secretion (Supplementary Fig. 6b, c). Altogether, these data suggest that CDC42-165aa regulates cell pyroptosis in an AIM2- or NLRP3-independent pathway. Considering that Pyrin inflammasome mediates pyroptosis by sensing the modification of Rho GTPases upon bacteria invasion[29], we hypothesized that circCDC42 regulates AM pyroptosis by promoting Pyrin inflammasome activation. Co-IP results showed that circCDC42 over-expression significantly increased the binding of pyrin and ASC in MH-S cells (Fig. 6a and Supplementary Fig. 6d). Colchicine, a specific Pyrin inflammasome inhibitor, was used to further elucidate whether CDC42-165aa-aggravated cell pyroptosis was mediated by Pyrin inflammasome activity. The results revealed that colchicine pretreatment dose-dependently reversed the viability of MH-S cells over-expressing CDC42-165aa after KP infection (Fig. 6b). In addition, ELISA results showed that after KP infection, colchicine blocked the effect of circCDC42 on promoting proinflammatory cytokine secretion (Fig. 6c). Immunofluorescence results also demonstrated that circCDC42 overexpression induced ASC speck formation in KP-infected MH-S cells, which was abolished by colchicine (Fig. 6d). Moreover, colchicine inhibited the expression of pyrin, CASP-1 cleavage, GSDMD cleavage, and IL-1β release induced by circCDC42 (Fig. 6e, f, and Supplementary Fig. 6e). Notably, CASP-1 knockdown inhibited the effect of CDC42-165aa on pyroptosis (Supplementary Fig. 6f, g), indicating that CASP-1 was indispensable for CDC42-165aa-induced pyroptosis. Collectively, these findings confirmed that CDC-165aa leads to the hyperactivation of Pyrin inflammasome, which aggravates pyroptosis. Moreover, we found that CDC42 knockdown inhibited the expression of Pyrin inflammasome, while CDC42-165aa

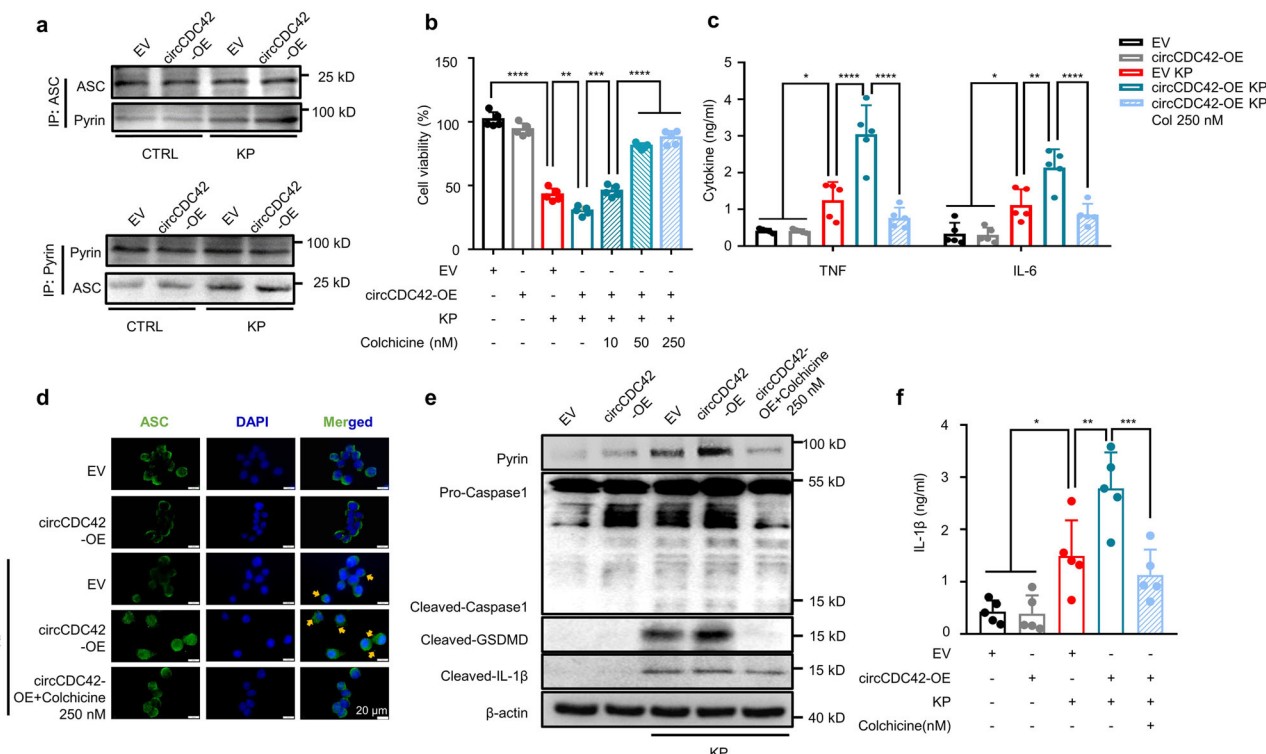

**Fig. 6 | CDC42-165aa promoted macrophage pyroptosis by activating the Pyrin inflammasome. a** MH-S cells were treated with empty vector (EV) or circCDC42 overexpression (OE) plasmid for 24 h and subsequently infected with KP at an MOI of 20:1 for 4 h. The binding of pyrin to ASC was detected by co-immunoprecipitation using the anti-Pyrin and ASC antibodies, respectively ($n = 3$ biological replicates). **b** MH-S cells were transfected with the indicated plasmid for 24 h, and treated with different concentrations of colchicine (left to right: 10, 50, and 250 nM) for 30 min before KP infection (MOI of 20:1) for 4 h. Cell viability was assessed by MTT assay ($n = 5$ biological replicates). **c** Expression levels of TNF and IL-6 in MH-S cells after KP treatment were measured by ELISA ($n = 5$ biological replicates). **d** MH-S cells were cells transfected with an indicated plasmid for 24 h, and treated with colchicine (250 nM) for 30 min before KP infection (MOI of 20:1 for 4 h). ASC speck formation (indicated by yellow arrows) was monitored by immunofluorescence ($n = 3$ biological replicates). Scale bar, 20 μm. **e** Expression of Pyrin, Caspase-1, GSDMD, and IL-1β in MH-S cells from the indicated groups was detected by immunoblotting ($n = 3$ biological replicates). **f** The secretion of IL-1β in the indicated MH-S cell supernatants was detected by ELISA ($n = 5$ biological replicates). Data are presented as means ± SD. **b, f** One-way ANOVA with Tukey test. **c** Two-way ANOVA with Bonferroni's test. *$p < 0.05$; **$p < 0.01$; ***$p < 0.001$; ****$p < 0.0001$. $p$-values: (**b**) $p = 2.19E\text{-}12$ (EV $vs$ EV + KP), $p = 0.0027$ (EV + KP $vs$ OE + KP), $p = 0.0002$ (OE + KP $vs$ OE + KP+Col10), $p = 1.83E\text{-}06$ (OE + KP+Col10 $vs$ OE + KP+Col50), $p = 2.15E\text{-}08$ (OE + KP+Col10 $vs$ OE + KP+Col250). (**c**) TNF: $p = 0.0164$ (EV vs. EV + KP), $p = 1.31E\text{-}07$ (EV + KP $vs$ OE + KP), $p = 3.48E\text{-}10$ (OE + KP $vs$ OE + KP+Col250); IL-6: $p = 0.0276$ (EV vs. EV + KP), $p = 0.0021$ (EV + KP $vs$ OE + KP), $p = 8.75E\text{-}05$ (OE + KP $vs$ OE + KP+Col250). See also Figure S6. Source data are provided as a Source data file.

overexpression partially restored Pyrin inflammasome expression (Supplementary Fig. 6h). This data revealed that the regulation of CDC42-165aa in Pyrin inflammasome-mediated pyroptosis is achieved through the parental gene *CDC42*. Compared with a previous study[20], we proposed that CDC42-165aa is involved in Pyrin inflammasome activation but alone is not sufficient for robust Pyrin inflammasome activation. The parent gene *CDC42* appears to be necessary for Pyrin inflammasome activation, and CDC42-165aa acts as a molecular switch in this process. Further studies are needed to unravel the intricate involvement of CDC42 and CDC42-165aa in the activation of inflammasomes, which will be explored in our follow-up studies dissecting the underlying mechanisms. The activation of inflammasome was also detected after PAO1 or *E. coli* infection (Supplementary Fig. 6i). Similar to the results obtained in KP infection, CDC-165aa increased the levels of Pyrin inflammasome in both PAO1 and *E. coli* infection. These results highlight the universality of CDC42-induced Pyrin inflammasome-mediated pyroptosis in Gram-negative infection.

## CircCDC42 triggers inflammation that aggravates mice lung injury after KP infection

To investigate the in vivo effect of CDC42-165aa, a KP-induced acute lung injury mouse model was constructed. CircCDC42 was specifically knocked down by targeting the mouse lung with a siRNA interference/ in vivo-jetpei hybrid delivery system, and then the circCDC42-siRNA or siNC control mice were intranasally challenged with $5 \times 10^6$ colony-forming units (CFU) of KP (Fig. 7a and Supplementary Fig. 7a). Mouse survival increased remarkably in the circCDC42 knockdown group post-KP infection (Fig. 7b). The bacterial burden of the lung and blood was dramatically decreased in the circCDC42 knockdown group compared with control mice (Fig. 7c). Lung histopathological examination was performed 24 h after KP infection to further assess lung injury. We identified significant histological alterations and inflammatory infiltration in the lungs of si-NC control mice 24 h after KP infection, while a slight lung injury was found in siCDC42 mouse lungs (Fig. 7d, e). To directly investigate the role of CDC42-165aa of AMs in vivo inflammatory response, primary AMs from mouse BALF were isolated, cultured, and expanded in a medium containing granulocyte-macrophage colony-stimulating factor (GM-CSF) for 14 days, as previously described[30–32]. After treatment with circCDC42- or siNC control siRNAs, AMs were adoptively transferred intratracheally (i.t.) into AM-depleted mouse lungs. Then, the mice were infected with the indicated CFU of KP (Supplementary Fig. 7b, c). It was found that the survival rate was substantially increased in circCDC42 knockdown mice within 72 h of KP infection (Supplementary Fig. 7d). Furthermore, siCDC42 treatment strongly inhibited the proinflammatory cytokines tumor necrosis factor (TNF) and IL-6 in mouse BALF (Supplementary Fig. 7e). We also observed increased recruitment of lymphocytes and

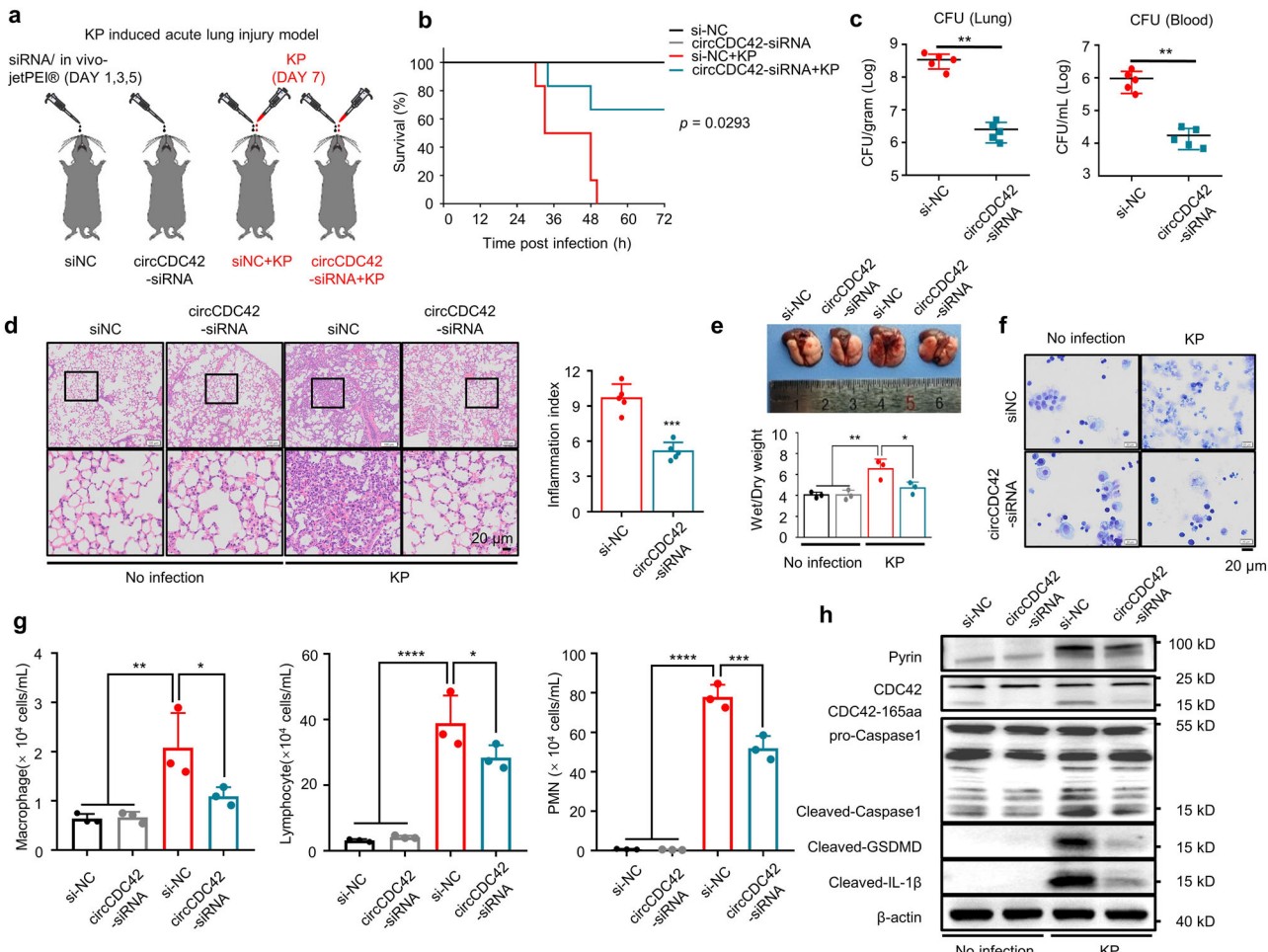

**Fig. 7 | circCDC42 Triggers Inflammation that Protects Mice from kp. a** The experimental procedures for the establishment of KP-induced mouse acute lung injury model and schematic time. Mice were intranasally injected with a vehicle containing either siNC or circCDC42-siRNA on Days 1, 3, and 5 to knock down circCDC42 in mice lungs. On Day 7, 50 µl KP suspensions (5 × 10⁶ CFU per mouse, $n = 12$ mice per group) was injected intranasally to establish an acute lung injury model. Mice in the control group received an equal volume of placebo suspension (normal saline), $n = 12$ mice per group. **b** Kaplan-Meier survival curves of KP-infected siNC-and circCDC42-siRNA–injected mice ($n = 6$, two independent experiments). Survival was recorded up to 72 h ($p = 0.0293$). **c** 24 hours after the intranasal infection of KP, blood and lung tissues of siNC and circCDC42-siRNA mice were collected to assess bacterial dissemination by CFU assay ($n = 5$ mice per group). **d** Left panels, lung injury, and inflammation were assessed by H&E staining of paraffin-embedded sections. Scale bar, 20 µm. Right panel, inflammatory cell infiltration was determined in the lung ($n = 5$ mice per group). **e** Upper panel, representative macroscopic picture of mice infected with KP or not. Lower panel, ratio of Wet/Dry weight were quantified ($n = 3$ mice per group). **f** AMs collected from WT donor mice were treated with circCDC42 siRNA or siNC for 48 h.

Clodronate liposomes were administrated to recipient mice to induce the depletion of AMs. Then, 0.8–1.0 × 10⁶ AMs resuspended in 100 µL PBS per receptor were administered via i.t. administration into AM-depleted recipient mice. Immune cells in BALF collected from KP-infected recipient mice were concentrated on glass slides, fixed, and stained by Quick-Diff. Representative images of BALF were recorded using a light microscope ($n = 3$ biological replicates). **g** Macrophage, lymphocytes, and PMN count in BALF from KP-infected siNC- and circCDC42 siRNA-AMs Recipient mice were determined through the Diff-Quick staining ($n = 3$ biological replicates). PMN, polymorphonuclear. **h** Expression of CDC42-165aa, Pyrin, Caspase-1, GSDMD and IL-1β in AMs isolated from BLAF was detected by immunoblotting ($n = 3$ biological replicates). Data are shown as the means ± SD. **b** Log-rank (Mantel-Cox) test. **c**, **d** Unpaired two-tailed $t$ test. **e**, **g** One-way ANOVA with Tukey test. *$p < 0.05$; **$p < 0.01$; ***$p < 0.001$; ****$p < 0.0001$. $p$-values: (**g**) Macrophage: $p = 0.0064$ (siNC vs. siNC + KP), $p = 0.0473$ (siNC + KP vs. si-1 + KP); Lymphocyte: $p = 3.46E-05$ (siNC vs. siNC + KP), $p = 0.0293$ (siNC + KP vs. si-1 + KP); PMN: $p = 4.22E-08$ (siNC vs. siNC + KP), $p = 0.0004$ (siNC + KP vs. si-1 + KP). See also Figure S7. Source data are provided as a Source data file.

polymorphonuclear leukocytes (PMNs) to WT mouse lungs, while circCDC42 knockdown mice relieved recruitment of lymphocytes and PMNs (Fig. 7f, g). In addition, circCDC42 knockdown inhibited the expression of pyrin, CASP-1 cleavage, GSDMD cleavage, and IL-1β release in AMs in mouse lungs (Fig. 7h and Supplementary Fig. 7f). These results consistently indicated that circCDC42 knockdown in AMs alleviates lung injury in mice after KP infection.

Overall, our data suggest that circCDC42/CDC42-165aa regulates Pyrin inflammasome-induced pyroptosis by competitively binding DOCK8. Pathogenic bacterial infection results in CDC42-165aa over-expression encoded by circCDC42 in the lungs, which competitively binds with CDC42 to DOCK8, thereby inhibiting its activation of

CDC42, ultimately aggravating Pyrin inflammasome activation and pyroptosis (Fig. 8).

## Discussion

Here, we present the functional characterization of circCDC42 stably expressed in the lung tissue, particularly in AMs. Our findings show that it participates in several biological functions following pathogen infection via its encoded protein, CDC42-165aa. The present study uncovered several important findings. First, circCDC42 expression was significantly increased in mouse lungs following bacterial infection(especially Gram-negative bacteria) leading to enhanced inflammatory responses, pyroptosis, lung injury, and mortality. This finding

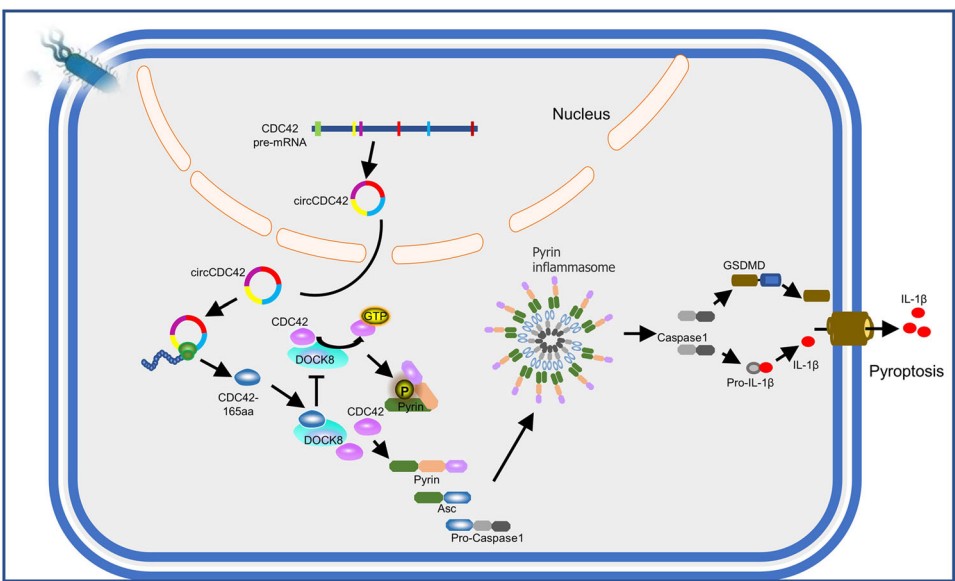

**Fig. 8 | Proposed model for the mechanism of action of circCDC42 in KP infection.** CircCDC42-encoded Cdc42-165aa was used as a protein bait to inhibit CDC42 activity through competitive binding to DOCK8, thereby activating the Pyrin/caspase-1/pro-IL-1β pathway and inducing alveolar macrophage pyroptosis.

reveals the important role of circCDC42 in regulating inflammatory responses during bacterial infection. Second, during bacterial infection, circCDC42 encodes a unique isoform of CDC42, namely CDC42-165aa, which further regulates Pyrin inflammasome-mediated pyroptosis by inhibiting the GTPase activity of CDC42. Third, elucidation of the functions of circCDC42 and its encoded protein CDC42-165aa provides insight into the function of circRNAs in regulating inflammatory responses and additional evidence that inhibition of circRNAs controls infection.

The current study focused on differentially expressed, translatable circRNAs in AMs after KP infection and identified circCDC42 as the most DEcR. Our data demonstrated that circCDC42 expression and function were tissue-specific and circCDC42 inhibition played a key role in the defense of the host lung against bacterial infection. Furthermore, we evaluated the coding potential of circCDC42 and found that circCDC42 contained a 536-nt ORF from the ATG start codon to the 12-nt TAA stop codon at the splice junction of the host gene. Our study demonstrated that circCDC42, in response to KP infection and driven by IRES, encoded a CDC42 isoform CDC42-165aa, which inhibited the biological function of CDC42 by competing for partner binding sites.

Previous studies summarized the activities of circRNAs such as microRNA sponges, protein decoys, and scaffolds[33–35]. Recent studies reported that unique circRNAs can be translated into proteins/peptides with biological functions[36–38]. Driven by IRES or modified by m6A, these circRNAs recruit ribosomes to initiate translation[7,39], suggesting that the translation of circRNAs plays a crucial role in the cellular stress response, apoptosis, and cell cycle regulation. The protein bait hypothesis has been proposed based on studies on the biological functions of circRNA-encoded proteins[40]. For example, some circRNA-encoded proteins, such as Mitogen-activated protein kinase 1 (MAPK1)-109aa encoded by circMAPK1 in gastric cancer[41], and SNF2 histone linker PHD RING helicase (SHPRH)-146aa encoded by circSHPRH in glioblastoma[42], inhibit the functions of their mRNA-encoded isoforms by competing for partner binding sites. Several previous studies have shown that circRNAs play a coding role in cancer, which leads to the hypothesis that imbalanced circRNA-encoded proteins can promote tumorigenesis. Nevertheless, it is unclear whether these proteins are involved in inflammatory responses, specifically in the case of pathogens. The present study identified a translatable circRNA and its

biological function in the innate immune defense against pathogen infection. Accordingly, circCDC42 may be a potential treatment target for bacterial infections, advancing our understanding of the roles of circRNAs in pathogen-host interactions.

The pyroptosis process is a type of proinflammatory cell death triggered by the activation of an inflammasome complex in response to microbial infection. During pathogen infection, moderate pyroptosis can recruit immune cells to clear pathogens. However, excessive pyroptosis not only causes cell death but also releases inflammatory factors to increase inflammatory responses, resulting in an immune imbalance. Therefore, pyroptosis might be a double-edged sword that is critical for immune defense. Although CDC42-165aa is not abundant in AMs, its concentration increases with the progression of infection accompanied by a significant increase in cell death. CDC42-165aa overexpression promoted the secretion of inflammatory cytokines and enhanced pyroptosis in KP-infected AMs. Our study indicates that CDC42-165aa is involved in KP-mediated AM pyroptosis and immunity. Several inflammasome activators have been shown to induce pyroptosis. For example, *Salmonella typhi* triggers pyroptosis by inducing the activation of the neuronal apoptosis inhibitory protein (NAIP)/NLRC4 inflammasomes in macrophages[43]. In addition, enterohemorrhagic *Escherichia coli* (EHEC) induces pyroptosis by activating the NLRP3 inflammasome in human macrophage THP-1 cells[44]. Unlike other inflammasomes, pyrin does not directly recognize or bind pathogenic microbial molecules; instead, it senses the modification and inactivation of Rho family small G proteins by various bacterial toxins in host cells and then assembles with the ASC to form a new inflammasome complex, which plays an important role in antibacterial innate immunity[45]. Our study demonstrated the activation of the Pyrin inflammasome by KP2044 (NTUH-K2044), a hypervirulent KP strain, which may lead to higher transmission and mortality. Our data delineate the mechanism underlying CDC42-165aa-mediated pyrin pyroptosis and the IL-1β secretion process and demonstrate the activation of the Pyrin inflammasome by KP2044, which may provide a mechanism to explain the increased pyroptosis that occurs after hypervirulent Kp infection. Consistent with several previous studies[12,13], the present study confirmed that NTUH-K2044 triggers cell death in macrophages displaying features of pyroptosis. However, a recent study found that hypervirulent KP strains do not induce pyroptosis compared with classical KP strains[46]. The discrepancy between

this study and ours may be due to differences in types of cells, bacterial infectious dosage, and time of infection. Future studies should assess the influence of infection conditions on KP-induced cell death.

We experimentally confirmed that both CDC42-165aa and its isoform CDC42 competed for binding sites with its partner DOCK8, thereby preventing the activation of CDC42, which may play an important role in activating the Pyrin inflammasome and increasing the pyroptosis of AMs. CDC42 is an important member of the Rho family of small G proteins, which function as molecular switches that are responsible for regulating the reorganization and formation of cytoskeletal proteins. They switch between active and inactive GDP-bound states by binding and hydrolyzing GEFs[47]. Modification and inactivation of the Rho family small G proteins are a common mechanism used by various pathogenic bacteria to control the host cytoskeleton and inhibit phagocytosis[48]. It has been previously suggested that the small Rho GTPase CDC42 is critical for pyrin activity and oligomerization of inflammasome complexes[18,20,49]. For example, *Burkholderia cenocepacia* inhibits the activity of Rho GTPase, and then pyrin senses the inactivation of Rho GTPase, induces the activation of CASP-1, and finally cleaves and activates GSDMD, thereby causing pyroptosis[50]. As a guardian of pyrin activation, the enzymatic activity of CDC42 promotes phosphorylation that inhibits inflammasome activation. Similarly, this study found that KP infection induced circCDC42 to express CDC42-165aa, which inactivated the CDC42 GTPase by competitively binding with its partner DOCK8, thereby promoting the activation of the Pyrin inflammasome and the secretion of IL-1β. However, treatment with a Pyrin inflammasome inhibitor colchicine gradually restored GSDMD cleavage and IL-1β secretion in a dose-dependent manner. Based on these data, it is believed that CDC42-165aa plays a vital role in Pyrin inflammasome hyperactivation. Compared with direct or indirect ways to regulate bacterial inflammasomes, this approach is unique and may lead to a new direction in the research of post-translational regulation of innate immunity via non-coding RNAs.

In summary, multi-molecule and cell-based approaches demonstrate the precise regulation of CDC42-165aa expression in mouse AMs to modulate inflammation. This study provides more insights into understanding the role of circRNA in immunity by treating bacterial infections.

### Limitations of the study
This study mainly presented its findings using cellular models. In vivo, we just performed some preliminary phenotypic evaluations to assess the role of CDC42-165aa in a KP-induced acute lung injury mouse model. Utilizing and developing appropriate animal or organoid models may provide a better understanding of how CDC42-165aa regulates Pyrin inflammasome function.

## Methods
### Mice and animal housing
C57BL/6 J mice were purchased from GemPharmatech Co., Ltd. (Nanjing, China) and kept in a temperature-controlled environment (22–25 °C) under a 12/12- h light/dark cycle. All mice were male and 7-8 weeks old. Animal experiments followed the ARRIVE (Animal Research: Reporting of In Vivo Experiments) guidelines. All protocols used in this study were approved by the Institutional Animal Care and Treatment Committee of Xuzhou Medical University (Approval No. 202306T017).

### Primary cells and cell lines
Mouse primary alveolar macrophages (AMs) were isolated and expanded[30,32]. Briefly, mice were euthanized, followed by dissection of their thoraxes and exposure of their tracheas. Lungs were lavaged with normal saline using a syringe with angiocath, and lavage fluid was retrieved into sterile tubes. AMs were isolated from bronchoalveolar lavage fluid and plated on 10-cm dishes in 10 mL of Roswell Park

Memorial Institute (RPMI) 1640 medium containing 10% fetal bovine serum (FBS), followed by the addition of the murine 5% granulocyte-macrophage colony-stimulating factor (GM-CSF) supernatant. AMs were passaged every 3-4 days and cultured for subsequent functional assays or transplantation studies. Mouse neutrophils (mononuclear histiocytic cell [MNHC]) and mouse CD4 + T cells were maintained in Dulbecco's Modified Eagle Medium (DMEM) with 10% FBS. Mouse natural killer (NK) cells were maintained in an NK92 special medium.

MH-S cells (murine AMs) were maintained in RPMI 1640 medium with 10% FBS. MLE-12 cells (murine type II lung epithelial cells) were maintained in DMEM/F12 medium with 10% FBS. Cells were cultured at 37 °C in 90% humidified air with 5% $CO_2$.

### Bacterial strains and growth condition
The *Klebsiella pneumoniae* strain (KP2044) was kindly provided by Dr. Min Wu (University of North Dakota), and the *Pseudomonas aeruginosa* strain (PAO1), *Staphylococcus aureus* (CPCC 160001), and *Escherichia coli* (ATCC 25922) were kept in our laboratory. The bacteria were grown in Luria-Bertani (LB) broth for 16 h at 37 °C with 220 rpm shaking.

### In vivo and in vitro bacterial infection
Bacteria were incubated overnight in LB broth at 37 ˚C with 220 rpm, centrifuged at 13400 × *g* for 5 min, and then resuspended in fresh LB broth for growth until the mid-log phase. Mammalian cells were treated with an antibiotic-free medium and then infected with bacteria in a multiplicity of infection (MOI) of 20:1 bacteria/cell[51,52]. Mice were anesthetized by intraperitoneal injection of Avertin (250 mg/kg) and intranasally infected with $5 \times 10^6$ colony-forming units (CFU) of bacterial strain in 50 μL phosphate-buffered saline (PBS)[53,54].

### Cell viability assay
Cell viability was determined using the MTT (3-[4,5-dimethylthiazol-2-yl]-2,5 diphenyl tetrazolium bromide) assay. Cells were plated on 96-well plates and incubated overnight. After attachment, cells were treated as above. Subsequently, cells were incubated in a medium (100 μL/well) with 50 μg of MTT for 4 h. Afterward, the MTT solution was removed, and 100 μL of dimethyl sulfoxide (DMSO) was added to dissolve MTT-formazan crystals. The absorbance was measured at 570 nm using a Multiskan FC Microplate Reader for quantification.

### Real-time quantitative PCR (RT-qPCR) analysis and primers
Total RNA was extracted using TRIzol (Life Technologies) according to the manufacturer's protocol. RNAs were reverse-transcribed by random primers using the HiScript II RT SuperMix (Vazyme, Nanjing, China), followed by RT-qPCR assay using the ChamQ SYBR qPCR Master Mix (Vazyme, Nanjing, China) with indicated primers. The relative gene expression was calculated using the $2^{-\Delta\Delta CT}$ cycle threshold method. Results were normalized to the glyceraldehyde 3-phosphate dehydrogenase (*GAPDH*) messenger RNA (mRNA) in each sample. All the primer sequences used are listed in Supplementary Table S1.

### RNA-seq analysis and identification of circRNA
RNA-seq and analysis were performed using Illumina Novaseq6000 by Gene Denovo Biotechnology Co., Ltd. (Guangzhou, China). Reads per kilobase per million mapped reads (RPKM) were applied to quantify circRNAs. circRNAs were blasted in the circBase for annotation. Sequences that could not be annotated were defined as novel circRNAs. Significant differentially expressed circRNAs (DEcRs) were identified as those with a fold change ≥ 2 and a *P*-value < 0.05 in a comparison between samples or groups.

### Actinomycin D assay
MH-S cells were seeded uniformly in 24-well plates. After 24 h of cultivation, cells were treated with 2 μg/mL of actinomycin D for 0, 2, 4, 8,

12 and 24 h, respectively. Next, total cellular RNA was extracted from the harvested cells. The relative expression of circRNA and mRNA was assessed using RT-qPCR, and the values were normalized to 0 h.

## RNase R treatment

Total RNA was extracted from the cells and incubated at 37 ˚C with or without RNase R (4 μ/mg, Epicenter Technologies, USA) for 30 min. RT-qPCR was performed to validate the stability of circRNA and the values were normalized to the *GAPDH* mRNA in RNase R-free sample.

## RNA subcellular isolation

Cytoplasmic and nuclear RNAs were isolated from MH-S cells using the RNA Subcellular Isolation Kit (Active Motif, USA) according to the manufacturer's instructions. Briefly, MH-S cells were lysed with a lysis buffer on ice. The supernatant was collected for cytoplasmic RNA extraction and the rest of the sediment was used for nuclear RNA. The extracted RNA was analyzed using RT-qPCR. The relative expression levels in the cytoplasm and nucleus were normalized to the internal reference genes *GAPDH* and *U6*, respectively.

## RNA fluorescence in situ hybridization (FISH)

Oligonucleotide probes with Cy3 labels specifically targeting the circCDC42 junction were designed and synthesized by GeneSeed (Guangzhou, China). MH-S cells were seeded on poly-L-lysine-coated coverslips (Sigma-Aldrich). The FISH assay was performed using a Fluorescence In Situ Hybridization kit (GeneSeed, Guangzhou, China) according to the manufacturer's guidelines. Confocal fluorescence images were obtained using a laser-scanning confocal microscope (Leica STELLARIS 5, Germany). The probe sequence is displayed in Supplementary Table S2.

## RNA interference and transfection

CircCDC42 and scramble small interfering RNAs (siRNAs) were obtained from Sangon Biotech (Shanghai, China). DOCK8 siRNA was purchased from Generay Biotech Co., Ltd. (Shanghai, China). JetPRIME transfection reagent (Polyplus) was used for siRNA transfection according to the manufacturer's guidelines. Target sequences are listed in Supplementary Table 1.

## Plasmid construction and transfection

*CircCDC42* overexpression (circCDC42-OE), circCDC42 ATG-delete (circCDC42 ATG-del), CDC42-165aa-3×Flag ORF plasmids were constructed and generated by chemical gene synthesis (Genechem, Shanghai, China) employing CMV enhancer-left circular frame-MCS-right circular frame-EF1a-ZsGreen1-SV40-puromycin as the backbone. Absent in melanoma 2 (AIM2), NLR family CARD domain-containing protein 4 (NRLC4), Caspase-1, and CDC42 short hairpin RNAs (shRNAs) were purchased from Generay Biotech Co., Ltd. (Shanghai, China). FuGENE® HD Transfection Reagent (Promega) or Lipofectamine 2000 reagent (Invitrogen) were used for cell transfection.

## Lactate dehydrogenase (LDH) release assay

LDH release was measured using the LDH Release Assay Kit (Solarbio) according to the manufacturer's protocols. A comparison of medium activity to cell lysate was used to calculate the percentage of LDH released from cells.

## Flow cytometry analysis

Flow cytometry was used to quantify cell death using the Annexin V-FITC/PI Detection kit (Beyotime) following the manufacturer's instructions. Briefly, cells were seeded in a 6-well plate and treated with indicated siRNAs or siRNA negative control (siNC) for 24–48 h. After infection with KP for 4 h, cells were harvested and stained with Annexin V-FITC and propidium iodide (PI) for 15 minutes. Cell death was analyzed using a flow cytometer (Thermo Fisher Scientific, USA).

## Immunoblotting analysis

Total protein was extracted using radioimmunoprecipitation assay (RIPA) lysis buffer (MedChemExpress, China) according to the manufacturer's protocol. The concentration of proteins was measured using bicinchoninic acid (BCA) assay kits. After sodium dodecyl-sulfate polyacrylamide gel electrophoresis (SDS-PAGE) separation, samples were transferred to nitrocellulose transfer membranes. The membranes were blocked with 5% skim milk in Tris-buffered saline with Tween 20 (TBST) buffer for 1 h, then incubated overnight with primary antibodies. After three washes with TBST buffer for 10 min, membranes were incubated at room temperature for 1 h with horseradish peroxidase (HRP)-labeled secondary antibodies. Protein signals were visualized using an enhanced chemiluminescence HRP substrate (MedChemExpress, China). The protein expression was quantified by ImageJ software. All antibodies used in this study are depicted in the Supplementary Table S1.

## Dual-luciferase assay systems

The wild-type or mutant IRES sequence of circCDC42 was constructed and inserted into the luciferase reporter gene. After 48 h of transfection, luciferase reporter assays were conducted utilizing the dual-luciferase reporter assay system (Promega, Madison, WI, USA). Relative luciferase activity was normalized to *Renilla* luciferase activity.

## Cdc42 Activation assay

Cdc42 Activation was performed using a Cdc42 Activation Pull-Down Assay Biochem Kit (Cytoskeleton, Inc., Denver, CO, USA) according to the manufacturer's guidelines. Approximately 500 μg of protein was incubated with PAK-PBD beads (10 μL) at 4 °C for 1 h on a rotator. After washing with 500 μL of wash buffer, the beads were centrifuged at 13400 × $g$ for 3 min at 4 °C. The beads were then boiled at 95 °C for 2 min in 20 μL of 2× laemmli sample buffer. Finally, samples were separated by SDS-PAGE and analyzed using western blot assay. The quantification of the number of proteins pulled down was normalized to the protein input.

## Immunofluorescence analysis

After being fixed in 4% paraformaldehyde (PFA) for 20 min, cells were washed three times in PBS and permeabilized for 10 min in PBS with 0.1% Triton X-100. Subsequently, cells were blocked with normal goat serum, incubated overnight with primary antibodies, and then counterstained with appropriate fluorescent secondary antibodies and 4',6-diamidino-2-phenylindole (DAPI). Fluorescence images were captured under a laser-scanning confocal microscope.

## ASC specks immunofluorescence

Cells were fixed with PFA. Following permeabilization with 0.1% Triton X-100 in PBS buffer, cells were stained with anti-ASC (Santa Cruz, sc-514414), CoraLite488-conjugated Goat Anti-Mouse IgG (Proteintech, #SA00013-1), and DAPI. ASC specks were visualized on a laser-scanning confocal microscope. Quantification was performed on 10 fields per sample.

## Microscopic observation of cell morphology

MH-S cells were seeded uniformly in 24-well plates. After 24 h, cells were treated with plasmid or siRNA. Bright-field images of live cells were captured under an Olympus IX71 microscope. The number of pyroptotic cells was calculated from 100 random cells under microscopic conditions by multiplying the number of pyroptotic cells by the total number of cells[55].

## Mass spectrum analysis

Protein samples were separated by SDS-PAGE. Then, target gel bands were cut and digested with trypsin (Promega, USA). Peptides were analyzed using a Q Exactive mass spectrometer (Thermo Fisher

Scientific, USA). Following mass spectrometry analysis, all the raw files were analyzed using pFind software (version 3.1.6).

## Mice models

For the siRNA transfection model, mice were intranasally administered the siRNA interference/in vivo-jetpei hybrid delivery system before bacterial infection following the manufacturer's instructions. Briefly, siNC (30 μM per mouse) or circCDC42 siRNA (30 μM per mouse) was dissolved in 5% glucose solution and transfection reagent in vivo jetPEI (Polyplus) according to the manufacturer's protocol. About 60 μL of the siRNA jetPEI complex was administered to mice via the nasal route on days 1, 3, and 5. Mice were anesthetized with Avertin and intranasally infected with KP at day 7 to establish an acute lung injury model ($n = 6$ mice/group).

## Histological and histomorphometric analyses

The lung wet/dry ratio was calculated as an index of lung injury (pulmonary edema)[12]. Mouse lungs were dissected and separated. The lower lobe of the right lung was measured to obtain the wet weight, then dried in an oven at 80 °C for 24 h to obtain the dry weight. The upper lobe of the right lung was weighed and homogenized for bacterial CFU measurements. The left lung was used for histological analysis. Lung tissues of three independent mice were immediately fixed in PFA, embedded in paraffin, and then cut into 4-μm sections. Hematoxylin and eosin (H&E) staining was performed to examine the morphology of the sections. The lung inflammation index was calculated on the following histologic features[56]: capillary hyperemia, hemorrhage, inflammatory cell infiltration, and alveolar wall thickness. A score of 0 denotes normal lungs; 1 denotes mild, < 25% lung involvement; 2 denotes moderate, 25–50% lung involvement; 3 denotes severe, 50–75% lung involvement; and 4 denotes very severe, >75% lung involvement. The total score was the sum of the above items.

## AM depletion and intratracheal transfer

Briefly, 50 μL of clodronate liposomes or control liposomes were administered directly into the lung via intratracheal (i.t.) administration twice every 24 h to deplete AMs[57,58]. Adoptive transfer of AMs was performed as previously reported[32,59]. AMs collected from WT donor mice were treated with circCDC42 siRNA or siNC for 48 h. Then, $0.8–1.0 \times 10^6$ AMs resuspended in 100 μL PBS per receptor were administered via i.t. administration into AM-depleted recipient mice. Recipient mice were anesthetized with Avertin and intranasally infected with KP for 24 hours to establish an acute lung injury model ($n = 6$ mice/group). The bronchoalveolar lavage fluid (BALF) was extracted using 0.8 mL of PBS, and total cells were counted using a hemocytometer. Cell smear was stained using the Diff-Quik stain kit (Solarbio) according to the manufacturer's instructions. Differential cells were counted under a microscope.

## Measurement of cytokines production

The cytokines produced in cell culture supernatants and mice serum were determined using an enzyme-linked immunosorbent assay (ELISA) kit. ELISA kits for mouse interleukin-6 (IL-6), tumor necrosis factor (TNF), and IL-1β were purchased from Proteintech Group, Inc. (Chicago, IL, USA).

## Statistics and reproducibility

Except for some animal studies, each experiment was repeated at least three times with similar results. GraphPad Prism version 7.0 was used for statistical analyses. Data are expressed as means ± standard deviation (SD). Data were compared using a one-way ANOVA with Bonferroni post-test or Mann-Whitney test. A two-tailed paired $t$ test was applied for paired samples. Survival analysis was performed using the Kaplan-Meier method and log-rank test for survival curves. $P < 0.05$ was considered statistically significant.

## Reporting summary

Further information on research design is available in the Nature Portfolio Reporting Summary linked to this article.

## Data availability

CircCDC42 was uploaded to the circBase database (http://www.circbase.org/). The RNA-seq analysis datasets generated during this study are available on the NCBl Sequence Read Archive under the Bioproject PRJNA984001 identifier. The protein mass spectrometry raw data have been deposited to the ProteomeXchange Consortium via the iProX partner repository[60,61] with the dataset identifier PXD045396. The remaining data are provided in the Article and Supplementary Information. Source data are provided with this paper.

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

## Acknowledgements

This study was supported by the National Natural Science Foundation of China (NSFC82270011), the Natural Science Foundation of Guizhou Provincial Scientific and Technology Department Grant (Grant No. 2022(626)), the Priority Academic Program Development of Jiangsu Higher Education Institutions (PAPD) and Postgraduate Research & Practice Innovation Program of Jiangsu Province (KYCX21_2579), the Key Research and Development Project of Xuzhou (KC22238). We sincerely thank Dr. Fuxing Dong from the Public Experimental Research Center of Xuzhou Medical University for his enthusiastic help in the experiment of laser scanning confocal microscopy.

## Author contributions

R. L. and S. H. conceived and designed the research. N. X., F. J., and G. D. performed in vitro experiments and interpreted the data. J. J., L. M., J. C. and M. W. performed and analyzed in vivo experiments. C. L. analyzed the RNA-seq data and provided critical tools. N. X. and R. L. wrote the manuscript. Y.S. reviewed and revised the manuscript.

## Competing interests

The authors declare no competing interests.
