## [Peer Review File · Nature Communications]

REVIEWER COMMENTS

Reviewer #1 (Remarks to the Author):

This is an interesting manuscript where, the authors investigate the role of CDC42-165aa, a protein encoded by circCDC42 in Klebsiella pneumonia. They show that CDC42-165aa is overexpressed in KP-infected alveolar macrophages. CDC42-165aa overexpression induced the hyperactivation of pyrin inflammasome, which aggravated alveolar macrophages pyroptosis, while the knockdown of circCDC42 attenuated lung injury in mice upon KP infection by inhibiting Pyrin inflammasome-mediated pyroptosis. They further show that CDC42-165aa stimulates Pyrin inflammasome by inhibiting CDC42 GTPase activation and could be a potential clinical target of KP infection. The manuscript provides novel insights into the role of CDC42 in K. pneumonia infection, which has not been investigated previously. Their data support that circCDC42 expression is increased in alveolar macrophages in KP infection, that encodes for CDC42-165aa which regulates pyroptosis.

However, the data does not conclusively show that the modulation in innate immune response in the in vivo model of Klebsiella pneumonia is entirely related to pyroptosis in alveolar macrophages. There are several other cell types that are at play in a pneumonia model, including neutrophil, lymphocytes and structural cells. Although the authors show that there is no activation in epithelial cells in vitro in a cell line, the other cell types contribute to the immune response. Furthermore there are other functions of CDC42-165aa which are not considered. There is no hypothesis of how bacterial factors contribute to the activation of circCDC42 in macrophages and in vivo. Although the results are interesting the approach and experimental design is not adequate to make conclusions that the authors have described.

Major comments:

- 1) The authors show that that circCDC42 expression is increased in alveolar macrophages in klebsiella infection, they also show that MLE 12 cells did not have an increased expression in vitro. However, they have not determined the expression of circCDC42 in other immune cells such as neutrophils which play a critical role in Klebsiella infection. In vivo, the immune response includes multiple cell types, which may contribute to the bacterial clearance and may have pyrin activation. Additional experiments to determine this are needed before a conclusion is made that CDC42 inhibition in alveolar macrophages alters the innate immune response.
- 2) In supplementary figure 1C, they show that that there is a significant increase in circCDC42 expression in lungs. Whether this is entirely from alveolar macrophages is unclear. Interestingly, the spleen also shows an increased expression (not sure if it is statistically significant), but this is likely from other cells such as lymphocytes

3) CDC42-165aa has functions other than regulation of pyroptosis, how this impacts the alveolar response is not considered or discussed.

4) The in vivo data in figure 7 shows circCDC42 siRNA improving survival. However, this inhibition is not macrophage specific and therefore it one cannot conclude that CDC42 in alveolar macrophage is the only regulator of pyroptosis in vivo

5) circCDC42 expression is also increased in other Gram negative infections, it will be interesting to investigate the bacterial factors that contribute to the increased expression.

Minor Comments:

1) In figure 1C where the authors show lungs from mice, do not adequately show differences, better images should be provided

2) Similarly in figure 1 e the histology differences are not representative enough to show the differences. It is unclear how the inflammation index was quantified. Cell counts from Broncho-alveolar lavage and cytokine measurements from BAL should be included to show inflammation.

Reviewer #2 (Remarks to the Author):

The study of Xu et al (NCOMMS-23-40201) investigates the role of a circular CDC42 mRNA, derived from linear CDC42 mRNA, that encodes for a truncated form of CDC42 with immunomodulatory functions in macrophages during *Klebsiella pneumoniae* (Kp) infection. It is an extensive study but with data mainly derived from an alveolar macrophage cell line (MH-S) and not primary cells, making it difficult to determine whether these findings are of importance. Moreover, the quality of several Western blots is insufficient to draw proper conclusions. Furthermore, information is lacking how experiments were performed. Therefore, it is difficult to determine whether the conclusions of this study are sound.

Major concerns:

1. The quality of a large number of blots is insufficient (Fig. 2D, 3A, 3C, 3E, 4A, 4F, 5C, 5E, 6E, 7G). Several blots appear to have artefacts, uneven staining/development as revealed by different background on different sides of the blots, non-specific bands, high background, etc. Moreover, several blots are cropped through critical bands. It is highly recommended to show the entire blots

(with the selection indicated) in the supplemental materials in order to assess the quality of the results.

It is unclear why the CDC42 blots in Figure 5F and G are in 2 pieces and not on the same blot?

Furthermore, since the polyclonal antibody used for analysis of CDC42 also recognizes RAC1, it is recommended to use a specific monoclonal antibody, not reactive with RAC1. Moreover, Santa Cruz SC-8401 antibody, which is specific for AA166-182 of CDC42 could be used to further proof that CDC42-165AA lacks the C-terminus of this protein.

Several blots of GSDMD are not convincing (Fig 3E, 4F, 6E, 7G). Additional blots with Cell Signalling antibody #34667 for cleaved GSDMD may further clarify this issue.

The rationale to show TNF and IL-6 on blot is unclear as these cytokines are rapidly secreted, in contrast to pro-IL-1beta. All data TNF and IL-6 data should be shown as ELISA result.

2. The proof that circCDC42 in primary (alveolar) macrophages regulates macrophage pyroptosis and the release of inflammatory cytokines in response to Kp is minimal and not convincing. Infact, the only proof that CDC42-165AA is expressed in primary macrophages is provided in Figure 2d, which shows just two samples from alveolar macrophages that were stimulated in vitro. These results require further substantiation; more samples and quantification. Moreover, it should also be shown that CDC42-165AA is expressed in primary macrophages during Kp-induced pneumonia (in vivo).

This analysis is necessary, since MH-S cells are immortalized macrophages with largely different characteristics as compared to terminally differentiated (non-dividing) primary alveolar macrophages.

Furthermore, the CDC42-165AA data from the mouse experiment also do not show any data on macrophages. Alveolar macrophages from siRNA treated mice should be analyzed (similar to Fig. 2d) to reveal that the treatment indeed reduced CDC42-165AA levels in these cells. In addition, alveolar macrophages from these siRNA-treated mice could analyzed for Kp-induced pyroptosis. Of note, the results shown in Fig. 7F/G and Fig. S7A-C could well be explained by differences in bacterial loads in the lung leading to differences in stimulation of the cells other than macrophages (endothelial cells, epithelial cells, neutrophils, monocytes, etc).

It is unclear whether expression of CDC42-165AA is restricted to alveolar macrophages, or is also produced by other macrophages, including Kupffer cells, spleen macrophages, peritoneal macrophages, bone marrow-derived macrophages? Since alveolar macrophages can only be obtained in small numbers from mice, it would be favorable to study these primary macrophages from other sites in higher numbers.

Finally, the conclusion that circCDC42 is primarily expressed in the lung (line 294) is incorrect, since other cell types in the lung have not been investigated.

3. It is unclear what the kinetics of CDC42-165AA expression are in MH-S cells after Kp stimulation, both on mRNA and protein level? Therefore, it is unclear whether the data shown CDC42-165AA mRNA and protein represent an optimal time point.

4. The effect of CDC42-165AA on pyroptosis is based on microscope pictures of a limited number of cells and unclear GSDMD blots. It is described in the text that pyroptotic cells were enumerated (line 530-535) but no data are shown of this analysis (rather than representative pictures). These results could be supplemented with measurement of LDH release. Furthermore, the Cell Signalling antibody #34667 for cleaved GSDMD could be used for immunofluorescent staining of pyroptotic cells. Moreover, over-expression of CDC42-165AA in macrophages lacking caspase-1 or pyrin would reveal whether CDC42-165AA is indeed crucial for Kp-induced pyroptosis induction.

5. The authors describe that they used KP2044 for infection of cells. Is this Klebsiella strain KP2044 identical to NTUH-K2044? NTUH-K2044 is a hypervirulent and hypermucoid K1:O1 strain that is highly resistant to phagocytosis. The authors describe that they infect the macrophages with KP2044, but this requires analysis of phagocytosis of this strain. Otherwise the text needs to be adjusted. In view of the resistance of NTUH-K2044 to phagocytosis, it should be explained (in the Discussion) why pyroptosis is important for host defense against this bacterial strain. A recent study that hypervirulent Kp strains do not induce pyroptosis in contrast to classical Kp strains might also be included in this discussion.

6. The Methods section needs adjustment:

- Viability assay with MTT is not described.

- Are bacteria not grown to log-phase? (Line 423) Or washed to remove LB?

- The methods for the mouse experiments are lacking; it is unclear how mice were treated with siRNA (Fig. 7A suggests intranasal treatment with siRNA, but in the legend for Fig. 7 it is described that siRNA was injected intravenously (line 809) as well as in the legend for Fig S7a); it is unclear how lung samples were generated; it is unclear how the Pathology index was generated; it is not described how many CFU were used for infection with bacterial strains other than Kp.

- In all figures it is unclear how many samples were analyzed? Line 428 it is described that all analyzes were done with n=6, but that appears not to be correct.

7. The legends of the figures require major adjustment since none provide essential information on the experiments, nature of the samples, timing, doses, number of replicates, etc, making it difficult to determine the value of the result in the particular figure.

8. Lastly, the suggestion of the authors that circCDC42 may serve as a biomarker of KP infection (Line 337) is far-fetched, to say the least, and should be omitted as this is not based on proper investigation.

Reviewer #3 (Remarks to the Author):

General Assessment:

The manuscript titled "Circular RNAs (circRNAs) and CDC42-165aa in Innate Immune Response Regulation" by Xu et al investigates the role of circCDC42 and its encoded protein CDC42-165aa in regulating innate immune responses during *Klebsiella pneumoniae* (KP) infection. The study proposes a potential mechanism by which CDC42-165aa stimulates the Pyrin inflammasome during KP infection. However, the study has several major issues and limitations, as outlined below.

Major Comments:

1. The manuscript lacks a comprehensive exploration of the specificity of circCDC42 induction. Given that other pathogens can induce circCDC42 expression, it is essential to investigate whether the activation of different inflammasomes, such as NLRC4 and Pyrin, is specific to KP or a more general response to bacterial infections or inflammatory ligands.
2. The authors should clarify why *E. coli* and *P. aeruginosa*, which induce circCDC42 in infected lungs, do not activate the Pyrin inflammasome. This discrepancy needs to be addressed.
3. Figure 2e raises questions about the role of circCDC42, as the levels of circCDC42 appear similar between overexpression and deletion. This inconsistency needs further validation.
4. The authors should emphasize in the introduction that the CDC42-DOCK8 and CDC42-pyrin signaling pathways have already been reported, to provide context and explain the novelty and significance of their study.
5. To provide more concrete evidence for the role of CDC42-165aa in regulating inflammasome activation, the authors should consider using knockout (KO) lines for various inflammasome components such as NLRP3, AIM2, NLRC4, Pyrin, NLRP1, and CASP11. This would help confirm or rule out the involvement of CDC42-165aa in regulating diverse inflammasome activation pathways.
6. Figure 3e exhibits inconsistent and unreliable data, as siNC+KP and EV+KP should show similar levels of inflammasome activation but do not. The authors should validate this data.
7. The observation that circCDC42 overexpression requires KP infection for inflammasome activation, as circCDC42 expression alone is not sufficient, should be reconciled with previous literature that describes KP-induced NLRP3 inflammasome activation. The authors should address this contradiction.

8. The manuscript should consider whether KP infection induces the activation of other cell death pathways, as this could be relevant to the study's context.

9. In Figure 7g, the inhibition of Pyrin expression by circCDC42 silencing raises questions about whether CDC42-165aa inhibits Pyrin expression itself rather than the assembly of the Pyrin inflammasome. This point needs further investigation.

10. Regarding Figures 5f and 5g, demonstrating the binding of DOCK8 and CDC42-165aa is insufficient to confirm that DOCK8 is crucial for inflammasome activation and cell death. The authors should consider providing DOCK8 knockout or knockdown data to support their claims.

11. The manuscript should address the potential ambivalent role of CDC42 in regulating pyrin inflammasome activity, as reported in a 2022 study. The authors should discuss whether CDC42-165aa can induce Pyrin inflammasome activation solely and consider the effects of interfering with CDC42 and overexpressing CDC42-165aa.

Minor Comments:

1. The resolution of images in Figures 3d and 4e is inadequate for observing dying cells undergoing pyroptosis. Improved image quality is necessary.

2. The manuscript should undergo thorough proofreading to correct grammatical and syntax errors, which can impact its readability and professionalism.

In conclusion, there are significant limitations, inconsistencies, and areas of concern that need to be addressed. The authors should provide additional data, clarification, and context to strengthen the manuscript's scientific rigor and impact.

Reviewer's comments

Reviewer #1:

This is an interesting manuscript where, the authors investigate the role of CDC42-165aa, a protein encoded by circCDC42 in Klebsiella pneumonia. They show that CDC42-165aa is overexpressed in KP-infected alveolar macrophages. CDC42-165aa overexpression induced the hyperactivation of pyrin inflammasome, which aggravated alveolar macrophages pyroptosis, while the knockdown of circCDC42 attenuated lung injury in mice upon KP infection by inhibiting Pyrin inflammasome-mediated pyroptosis. They further show that CDC42-165aa stimulates Pyrin inflammasome by inhibiting CDC42 GTPase activation and could be a potential clinical target of KP infection. The manuscript provides novel insights into the role of CDC42 in K. pneumonia infection, which has not been investigated previously. Their data support that circCDC42 expression is increased in alveolar macrophages in KP infection, that encodes for CDC42-165aa which regulates pyroptosis.

However, the data is does not conclusively show that the modulation in innate immune response in the in vivo model of Klebsiella pneumonia is entirely related to pyroptosis in alveolar macrophages. There are several other cell types that are at play in a pneumonia model, including neutrophil, lymphocytes and structural cells. Although the authors show that there is no activation in epithelial cells in vitro in a cell line, the other cell types contribute to the immune response. Furthermore there are other functions of CDC42-165-aa which are not considered. There is no hypothesis of how bacterial factors contribute to the activation of circCD42 in macrophages and in vivo. Although the results are interesting the approach and experimental design is not adequate to make conclusions that the authors have described.

Response: Thank you very much for your helpful suggestions. We carefully read them and actively addressed each of your comments.

Major comments:

1) *The authors show that that circCDC42 expression is increased in alveolar macrophages in klebsiella infection, they also show that MLE 12 cells did not have an increased expression in vitro. However, they have not determined the expression of circCDC42 in other immune cells such as neutrophils which play a critical role in Klebsiella infection. In vivo, the immune response includes multiple cell types, which may contribute to the bacterial clearance and may have pyrin activation. Additional experiments to determine this are needed before a conclusion is made that CDC42 inhibition in alveolar macrophages alters the innate immune response.*

Response: Thank you very much for your helpful suggestions. In the revised manuscript, we detected the expression of circCDC42 in most major types of alveolar immune cells, including neutrophils, lymphocytes, NK cells and alveolar macrophages by qRT-PCR analysis (Figure 1j). We found that circCDC42 had the highest abundance in alveolar macrophages (Figure 1j, line 133-136). In addition, we observed the localization of circCDC42 in lung sections by fluorescence in situ hybridization, and the results further indicated that circCDC42 was more abundant in AMs (see Figure 1k).

2) *In supplementary figure 1C, they show that that there is a significant increase in circCDC42 expression in lungs. Whether this is entirely from alveolar macrophages is unclear. Interestingly, the spleen also shows an increased expression (not sure if it is statistically significant), but this is likely from other cells such as lymphocytes*

Response: We found that circCDC42 was mainly expressed in lungs, followed by spleens (Figure S1d). However, the increase of circCDC42 expression was more significantly in lungs after KP infection, compared to other tissues (Figure S1e, line 131-133). Furthermore, RT-qPCR data and FISH data together revealed that circCDC42 was mainly expressed in AMs, instead of other immune cells, such as neutrophils, NK cells and T cells (Figure 1j and 1k). These data suggested that circCDC42 play dominant roles in AMs of lung tissues. This manuscript we focused

on the role of circCDC42 in lungs, and we will in the future study the function of circCDC42 in spleens, as you suggested.

3) CDC42-165aa has functions other than regulation of pyroptosis, how this impacts the alveolar response is not considered or discussed.

Response: Thank you very much for your helpful suggestions. In order to detect the main cell death type caused by high level of circCDC42 expression, we performed the flow cytometry analysis on KP-infected alveolar macrophages. We found that PI-positive cells accounted for the vast majority of KP-infected AMs, while Annexin-V+/PI- cells accounted for only a small fraction (Figure S3d, line 199-202). Since PI positive is generally considered to be the main marker of pyroptosis^{2,4}, we believed CDC42-165aa has functions mainly on cell pyroptosis. KEGG data also supported the important role of CDC42-165aa on cell pyroptosis (Figure S3f, line 202-205), thus we focused on cell pyroptosis in this study, but we will determine other functions of CDC42-165aa in the future, as you suggested.

4) The in vivo data in figure 7 shows circCDC42 siRNA improving survival. However, this inhibition is not macrophage specific and therefore it one cannot conclude that CDC42 in alveolar macrophage is the only regulator of pyroptosis in vivo.

Response: Thank you very much for your helpful suggestions. In the revised manuscript, we isolated primary AMs from BALF of wildtype C57 mice, and then treated these primary AMs with circCDC42 or siNC siRNA to knock down circCDC42 expression^{5,6}. After the confirmation of stable low circCDC42 expression, we transferred these AMs to macrophage-depleted mouse lungs (line 334-340). Data showed that knockdown of circCDC42 significantly improved the survival of the mice after KP infection (Figure S7b and S7c, line 340-341). This in vivo result determined that circCDC42 in AMs is the most important regulator of pyroptosis.

5) circCDC42 expression is also increased in other Gram negative infections, it will be interesting to investigate the bacterial factors that contribute to the increased

expression.

Response: Thank you very much for your helpful suggestions. Here we reported that circCDC42 expression is all increased in mouse lungs during multiple Gram negative infections, including *K. pneumoniae*, *Escherichia coli*, and *P. aeruginosa* PAO1 (Figure S1g, line 137-141). This result suggested that the role of circCDC42 may extend to a broad range of Gram-negative bacteria. In this study, we are mainly concerned about KP infection, because according to the Annual report of China Antimicrobial Surveillance Network (CHINET-2022), KP is the most common Gram-negative pathogens isolated in clinics.

Minor Comments:

1) In figure 1C where the authors show lungs from mice, do not adequately show differences, better images should be provided

Response: We think this comment may refer to Figure 7C. We replaced it with better images (now Figure 7e) and further quantified the data in the revised version.

2) Similarly in figure 1e the histology differences are not representative enough to show the differences. It is unclear how the inflammation index was quantified. Cell counts from Broncho-alveolar lavage and cytokine measurements from BAL should be included to show inflammation.

Response: According to your suggestion, we have replaced the histology data with better images (now Figure 7d). In the section of Experimental Methods, we described in detail how the inflammation index was calculated (line 661-667). We also added the data of cell counts and cytokine measurements from Broncho-alveolar lavage in the revised manuscript (Figure 7g and S7e, line 342-346).

Reviewer #2:

The study of Xu et al (NCOMMS-23-40201) investigates the role of a circular CDC42 mRNA, derived from linear CDC42 mRNA, that encodes for a truncated form of CDC42 with immunomodulatory functions in macrophages during Klebsiella pneumoniae (Kp) infection. It is an extensive study but with data mainly derived from an alveolar macrophage cell line (MH-S) and not primary cells, making it difficult to determine whether these findings are of importance. Moreover, the quality of several Western blots is insufficient to draw proper conclusions. Furthermore, information is lacking how experiments were performed. Therefore, it is difficult to determine whether the conclusions of this study are sound.

Response: Thank you for taking the time to read our manuscript and giving us valuable reviews and suggestions. We have studied them carefully and addressed your comments point-by-point.

Major concerns:

1) The quality of a large number of blots is insufficient (Fig. 2D, 3A, 3C, 3E, 4A, 4F, 5C, 5E, 6E, 7G). Several blots appear to have artefacts, uneven staining/development as revealed by different background on different sides of the blots, non-specific bands, high background, etc. Moreover, several blots are cropped through critical bands. It is highly recommended to show the entire blots (with the selection indicated) in the supplemental materials in order to assess the quality of the results. It is unclear why the CDC42 blots in Figure 5F and G are in 2 pieces and not on the same blot?

Response: According to your suggestion, we improved the quality of blots and provided the entire uncutted blots (with the selection indicated) in the supplemental materials (source data file). We also replaced new blots for Figure 5F and G, making sure that full length CDC42 and CDC42-165aa were in the same blot.

Furthermore, since the polyclonal antibody used for analysis of CDC42 also recognizes RAC1, it is recommended to use a specific monoclonal antibody, not

reactive with RAC1. Moreover, Santa Cruz SC-8401 antibody, which is specific for AA166-182 of CDC42 could be used to further proof that CDC42-165AA lacks the C-terminus of this protein.

Response: The CDC42 monoclonal antibody mostly target the full sequence of CDC42, can't simultaneously match CDC42-165aa. The apparent molecular weights of CDC42 (191aa) and RAC1(192aa) are 21-23 KD, both greater than CDC42-165aa predicted molecular weights 17 KD, so the polyclonal antibody used in this study did not influence the observation of CDC42-165aa blotting.

Moreover, in the revised manuscript, we used the Santa Cruz SC-8401 antibody to detect full length CDC42 and CDC42-165aa. Only full length CDC42 was detected by this antibody, demonstrating that CDC42-165aa lacks the C-terminus (Figure S2f, line 179-181).

Several blots of GSDMD are not convincing (Fig 3E, 4F, 6E, 7G). Additional blots with Cell Signalling antibody #34667 for cleaved GSDMD may further clarify this issue.

Response: In the revised manuscript, we used the Cell Signalling antibody #34667 to detect GSDMD, as you suggest. Data are consistent with previous results, demonstrating that circCDC42 plays critical role on GSDMD activities (Figure 3f, 4e, 6e, 7h).

The rationale to show TNF and IL-6 on blot is unclear as these cytokines are rapidly secreted, in contrast to pro-IL-1beta. All data TNF and IL-6 data should be shown as ELISA result.

Response: According to your suggestion, we added ELISA data of TNF and IL-6 in the revised manuscript (Figure 3c, 6c, s7e).

2) The proof that circCDC42 in primary (alveolar) macrophages regulates macrophage pyroptosis and the release of inflammatory cytokines in response to Kp is minimal and not convincing. Infact, the only proof that CDC42-165AA is expressed in

primary macrophages is provided in Figure 2d, which shows just two samples from alveolar macrophages that were stimulated in vitro. These results require further substantiation; more samples and quantification. Moreover, it should also be shown that CDC42-165AA is expressed in primary macrophages during Kp-induced pneumonia (in vivo).

This analysis is necessary, since MH-S cells are immortalized macrophages with largely different characteristics as compared to terminally differentiated (non-dividing) primary alveolar macrophages.

Response: Thank you for your valuable comments on our manuscript which we have amended according to your suggestions. In the revised manuscript, we isolated primary AMs from BALF of mice, and cultured in GM-CSF-medium as previously described^[1, 2]. We examined the expression of CDC42-165aa in primary alveolar macrophages (AMs) at different time points after KP infection. As shown in Figures S2f (right panel), the protein level of CDC42-165aa peaked at 4-8 hours (Figures S2e and S2f). We also detected high expression of CDC42-165aa in AMs isolated from BALF of mice with KP infection for 24h (Figures s2h).

Furthermore, the CDC42-165AA data from the mouse experiment also do not show any data on macrophages. Alveolar macrophages from siRNA treated mice should be analyzed (similar to Fig. 2d) to reveal that the treatment indeed reduced CDC42-165AA levels in these cells. In addition, alveolar macrophages from these siRNA-treated mice could analyzed for Kp-induced pyroptosis. Of note, the results shown in Fig. 7F/G and Fig. S7A-C could well be explained by differences in bacterial loads in the lung leading to differences in stimulation of the cells other than macrophages (endothelial cells, epithelial cells, neutrophils, monocytes, etc).

Response: In the revised manuscript, we isolated primary AMs from BALF of wildtype C57 mice, and the treated these primary AMs with circCDC42 or siNC

siRNA to knock down circCDC42 expression^{5,6}. After the confirmation of stable low circCDC42 expression, we transferred these AMs to macrophage-depleted mouse lungs (line 334-340). Data showed that knockdown of circCDC42 significantly improved the survival of the mice after KP infection (Figure S7b and S7c, line 340-341). We also found that the expression of CDC42-165aa and pyroptosis-related proteins in AMs were inhibited by knocking down circCDC42 (Figures 7h, line 346-348). These in vivo results determined that circCDC42 in AMs is the most important regulator of pyroptosis.

It is unclear whether expression of CDC42-165AA is restricted to alveolar macrophages, or is also produced by other macrophages, including Kupffer cells, spleen macrophages, peritoneal macrophages, bone marrow-derived macrophages? Since alveolar macrophages can only be obtained in small numbers from mice, it would be favorable to study these primary macrophages from other sites in higher numbers.

Response: We found that circCDC42 was mainly expressed in lungs, followed by spleens (Figure S1d.). However, the increase of circCDC42 expression was more significantly in lungs after KP infection, compared to other tissues (Figure S1e, line 131-133). This manuscript we focused on the role of circCDC42 in alveolar macrophages, and we will in the future study the function of circCDC42 in Kupffer cells, spleen macrophages, peritoneal macrophages, bone marrow-derived macrophages, as you suggested.

Finally, the conclusion that circCDC42 is primarily expressed in the lung (line 294) is incorrect, since other cell types in the lung have not been investigated.

Response: In the revised manuscript, we detected the expression of circCDC42 in most major types of alveolar immune cells, including neutrophils, lymphocytes, NK cells and alveolar macrophages by qRT-PCR analysis (Figure 1j). We found that circCDC42 had the highest abundance in alveolar macrophages (Figure 1j, line 133-136). In addition, we observed the localization of circCDC42 in lung sections by

fluorescence in situ hybridization, and the results further indicated that circCDC42 was more abundant in Ams (see Figure 1k, line 133-136).

3) It is unclear what the kinetics of CDC42-165AA expression are in MH-S cells after KP stimulation, both on mRNA and protein level? Therefore, it is unclear whether the data shown CDC42-165AA mRNA and protein represent an optimal time point.

Response: Thank you for your very helpful suggestions. We detected CDC42-165aa on mRNA and protein levels at different time points. We found that circCDC42 RNA and protein levels peaked at 4-8 hours infection with KP (shown in Figures S2e and S2f), while the cell viability were significantly decreased at 4 hours (Figures S2g). Therefore, we selected 4 hours for further experiments, focused on the biological function of circCDC42 RNA and CDC42-165AA after 4 hours of KP infection.

4) The effect of CDC42-165AA on pyroptosis is based on microscope pictures of a limited number of cells and unclear GSDMD blots. It is described in the text that pyroptotic cells were enumerated (line 530-535) but no data are shown of this analysis (rather than representative pictures). These results could be supplemented with measurement of LDH release. Furthermore, the Cell Signalling antibody #34667 for cleaved GSDMD could be used for immunofluorescent staining of pyroptotic cells. Moreover, over-expression of CDC42-165AA in macrophages lacking caspase-1 or pyrin would reveal whether CDC42-165AA is indeed crucial for Kp-induced pyroptosis induction.

Response: Thank you for your valuable comments. We replaced the Figure 3d and 4c with better images and pyroptotic cells were further quantified in the revised manuscript as previously described ⁷ (Figure S3g and S4c). In addition, according to your suggestion, we provide LDH release experiment as a further supplementary evidence (Figure 3e and 4d, line 212-215). Since Cell signaling antibody #34667 is not suitable for immunofluorescence (according to the antibody's specification), we used Cell signaling antibody #34667 for western blot detection of cleaved GSDMD.

Moreover, we used Caspase-1 shRNA to knock down Caspase-1 expression in

different CDC42-165aa level MH-S cells. Results showed that overexpression of CDC42-165aa in macrophages lacking caspase-1 inhibited KP-induced cell pyroptosis and LDH release (Figures S6f and S6g, line 300-302). We also used the Pyrin inhibitor colchicine in our experiment and showed similar results. Colchicine blocked the effect of circCDC42 on promoting cell pyroptosis, LDH release and pyroptosis-related protein expression (Figures 6b-6f, and line 290-300). These findings collectively confirmed that CDC-165aa lead to the hyperactivation of pyrin inflammasome, which aggravates pyroptosis.

5) The authors describe that they used KP2044 for infection of cells. Is this Klebsiella strain KP2044 identical to NTUH-K2044? NTUH-K2044 is a hypervirulent and hypermucoid K1:O1 strain that is highly resistant to phagocytosis. The authors describe that they infect the macrophages with KP2044, but this requires analysis of phagocytosis of this strain. Otherwise the text needs to be adjusted. In view of the resistance of NTUH-K2044 to phagocytosis, it should be explained (in the Discussion) why pyroptosis is important for host defense against this bacterial strain. A recent study that hypervirulent Kp strains do not induce pyroptosis in contrast to classical Kp strains might also be included in this discussion.

Response: We thank the reviewer for this comment. The *Klebsiella* strain KP2044 we used in this study is indeed NTUH-K2044. Although it is generally believed that highly virulent KP is resistant to phagocytosis, our results in DIFF staining from BALF, confirm that KP2044 can be phagocytosed by alveolar macrophages (Figures 7f). This may be related to the load of bacteria and cell status.

Actually, several studies confirmed that KP2044 can be phagocytosed and survive in macrophages without being killed^{8,9}. This characteristic is similar to that of intracellular bacteria that induce pyroptosis. A recent study found that hypervirulent KP strains do not induce pyroptosis in contrast to classical Kp strains. However, in our study we show that KP2044 triggers cell death in macrophages displaying features

of pyroptosis. This result is consistent with several other previous studies of the induced upon NTUH-K2044 infection^{10,11}. These apparently contradictory may be related with the different bacterial dosage and time of infection. We perform pyroptosis assays using MH-S or primary AMs infected with NTUH-K2044 at a multiplicity of infection (MOI) of 20 for 4 h, whereas the previous report used BMDMs or RAW264.7 cells infected with the different strains at MOI of 10 for 90 min. We speculate that the induction of pyroptosis may be delayed in hvKp-infected macrophages. However, further studies are needed to identify the factors that cause these differences in host cells.

According to your valuable suggestion, we added that to the Discussion section.

6. The Methods section needs adjustment:

- Viability assay with MTT is not described.

Response: We have added the viability assay with MTT in the Method section (line 511-518).

- Are bacteria not grown to log-phase? (Line 423) Or washed to remove LB?

Response: Bacteria were were grown in LB broth at 37°C with 220 rpm shaking overnight, following by centrifugeing at 12000 rpm for 5 min, and then resuspended in 5 mL fresh LB broth to grow until the mid-log phase. We have added this description in the Method section (line 504-507).

- The methods for the mouse experiments are lacking; it is unclear how mice were treated with siRNA (Fig. 7A suggests intranasal treatment with siRNA, but in the legend for Fig. 7 it is described that siRNA was injected intravenously (line 809) as well as in the legend for Fig S7a); it is unclear how lung samples were generated; it is unclear how the Pathology index was generated; it is not described how many CFU

were used for infection with bacterial strains other than *Kp*.

Response: Sorry for our carelessness. The siRNA was injected intranasally to mice. We have corrected it to our revised manuscript. In the revised manuscript, we detailly described the mouse use in Method section, including lung samples, Pathology index, et al. *In vitro*, 5×10^6 CFU of KP were used for infection with bacterial strains as previous described^{12,13} (line 508-510).

- *In all figures it is unclear how many samples were analyzed? Line 428 it is described that all analyzes were done with n=6, but that appears not to be correct.*

Response: We apologize for the lack of clarity regarding the number of used mice in the previous manuscript. In the revised version, we clearly figured out the number of mice used in each experiment, as also indicated in the Figure legends (line 864-1043).

7) *The legends of the figures require major adjustment since none provide essential information on the experiments, nature of the samples, timing, doses, number of replicates, etc, making it difficult to determine the value of the result in the particular figure.*

Response: Thank you very much for your valuable comments. We have reviewed and added these key details to the Figure legends to improve the accuracy of the manuscript (line 864-1043).

8) *Lastly, the suggestion of the authors that circCDC42 may serve as a biomarker of KP infection (Line 337) is far-fetched, to say the least, and should be omitted as this is not based on proper investigation.*

Response: We removed the description that circCDC42 was a biomarker in the revised manuscript.

Reviewer #3:

General Assessment:

The manuscript titled "Circular RNAs (circRNAs) and CDC42-165aa in Innate Immune Response Regulation" by Xu et al investigates the role of circCDC42 and its encoded protein CDC42-165aa in regulating innate immune responses during Klebsiella pneumoniae (KP) infection. The study proposes a potential mechanism by which CDC42-165aa stimulates the Pyrin inflammasome during KP infection. However, the study has several major issues and limitations, as outlined below.

Major Comments:

1) *The manuscript lacks a comprehensive exploration of the specificity of circCDC42 induction. Given that other pathogens can induce circCDC42 expression, it is essential to investigate whether the activation of different inflammasomes, such as NLRC4 and Pyrin, is specific to KP or a more general response to bacterial infections or inflammatory ligands.*

Response: Thank you very much for your valuable comments. We conformed that circCDC42 expression is all increased in mouse lungs during multiple Gram negative infections, including *K. pneumoniae*, *Escherichia coli*, and *P. aeruginosa* PAO1 (Figure S1g, line 137-141). This result suggested that the role of circCDC42 may extend to a broad range of Gram-negative bacteria. To investigate which inflammasomes associated with pyroptosis CDC42-165aa affects, we assessed the expression of AIM2, NLRC4, NLRP3 and Pyrin, four inflammasome signature proteins in MH-S cells. Immunoblotting data showed that overexpression of CDC42-165aa increased the Pyrin expression in MH-S cells upon KP infection. Of note, although NLRP3 expression was increased in KP infection, its expression was unaffected through overexpression of CDC42-165aa. (Figures S6a). In addition, no increase in NLRC4 and AIM2 expression was observed (Figures S6a). ELISA data also showed that knockdown of AIM2 or NLRP3 did not affect IL-1 β secretion induced by CDC42-165aa (Figures S6b and S6c). These results suggesting that

CDC42-165aa regulated cell pyroptosis in an AIM2- or NLRP3-independent pathway. Additionally, we measured the expression of other inflammasomes upon bacterial infection. We observed similar results in Pyrin expression (Figure S6f). The results showed that overexpression of CDC42-165aa increased the Pyrin expression in MH-S cells upon PAO1 and E.coli infection, while produced no effect on AIM2, NLRC4, NLRP3 protein level. In this study, we mainly focused on KP infection, because according to the Annual report of China Antimicrobial Surveillance Network (CHINET-2022), KP is the most common Gram-negative pathogens isolated in clinics.

2) *The authors should clarify why E. coli and P. aeruginosa, which induce circCDC42 in infected lungs, do not activate the Pyrin inflammasome. This discrepancy needs to be addressed.*

Response: Thank you very much for your valuable comments. In the revised manuscript, we determined that *E. coli* and *P. aeruginosa* also induce circCDC42 in infected lungs, thereby activating the Pyrin inflammasome (Figure S6i, line 314-319). These findings reveal that circCDC42 may play a similar role in regulating Pyrin inflammasome signaling pathway during bacterial infection.

3) *Figure 2e raises questions about the role of circCDC42, as the levels of circCDC42 appear similar between overexpression and deletion. This inconsistency needs further validation.*

Response: In Figure 2e, circCDC42 ATG delete plasmid just remove the start codon "ATG" from circCDC42 ORF, and other sequences are the same as circCDC42 overexpression plasmid. So, the levels of circCDC42 is similar, but the ability of encode protein is inhibited.

4) *The authors should emphasize in the introduction that the CDC42-DOCK8 and CDC42-pyrin signaling pathways have already been reported, to provide context and explain the novelty and significance of their study.*

Response: Thank you very much for your valuable comments. In the revised manuscript, we have added the introduction of CDC42-DOCK8 and CDC42-pyrin signaling pathways, and emphasized the novelty and significance of their study, by comparing our findings with previous works (line 74-85).

5) To provide more concrete evidence for the role of CDC42-165aa in regulating inflammasome activation, the authors should consider using knockout (KO) lines for various inflammasome components such as NLRP3, AIM2, NLRC4, Pyrin, NLRP1, and CASP11. This would help confirm or rule out the involvement of CDC42-165aa in regulating diverse inflammasome activation pathways.

Response: Thank you for your valuable suggestions. According to your advice, We used shRNA to knock down AIM2, NLRP3, or Caspase1 expression in MH-S cells, respectively. We observed that the IL-1 β release were in similar levels in AIM2 or NLRP3 knockdown MH-S cells, compared to wildtype cells (Figures S6b and S6c, line 282-284). We also used the Pyrin inhibitor colchicine in our experiment and showed similar results. Colchicine blocked the effect of circCDC42 on promoting cell pyroptosis, LDH release and pyroptosis-related protein expression. (Figures 6b-6f, and line 290-300). Moreover, we used Caspase-1 shRNA to knock down Caspase-1 expression in different CDC42-165aa level MH-S cells. Results showed that overexpression of CDC42-165aa in macrophages lacking caspase-1 inhibited KP-induced cell pyroptosis and LDH release (Figures S6f and S6g, line 300-302). These findings collectively confirmed that CDC-165aa lead to the hyperactivation of pyrin inflammasome, which aggravates pyroptosis.

6) Figure 3e exhibits inconsistent and unreliable data, as siNC+KP and EV+KP should show similar levels of inflammasome activation but do not. The authors should validate this data.

Response: We are thankful to the reviewer for pointing out this oversight. This error has been corrected in the revised manuscript ((Figures 3f and S3h).

7) *The observation that circCDC42 overexpression requires KP infection for inflammasome activation, as circCDC42 expression alone is not sufficient, should be reconciled with previous literature that describes KP-induced NLRP3 inflammasome activation. The authors should address this contradiction.*

Response: Previous literature describes that KP induced NLRP3 inflammasome activation^{10,14}. However, in this study, we reported that KP may not only induce the activation of the NLRP3 inflammasome, but also the Pysin inflammasome, suggesting that multiple inflammasome signals are involved in the response to KP infection. Here, we proposed that circCDC42 regulates cell pyroptosis mainly through the Pysin inflammasome pathway (lines 282-300).

8) *The manuscript should consider whether KP infection induces the activation of other cell death pathways, as this could be relevant to the study's context.*

Response: Thank you very much for your helpful suggestions. In order to detect the main cell death type caused by high level of circCDC42 expression, we performed the flow cytometry analysis on KP-infected alveolar macrophages. We found that PI-positive cells accounted for the vast majority of KP-infected AMs, while Annexin-V+/PI- cells accounted for only a small fraction (Figure S3d, line 195-201). Since PI positive is generally considered to be the main marker of pyroptosis²⁻⁴, we believed CDC42-165aa has functions mainly on cell pyroptosis. KEGG data also supported the important role of CDC42-165aa on cell pyroptosis (Figures S3f, line 202-205), thus we focused on cell pyroptosis in this study, but we will determine other functions of CDC42-165aa in the future, as you suggested.

9) *In Figure 7g, the inhibition of Pysin expression by circCDC42 silencing raises questions about whether CDC42-165aa inhibits Pysin expression itself rather than the assembly of the Pysin inflammasome. This point needs further investigation.*

Response: Thank you for your suggestion. Immunoblotting data showed that high level of circCDC42 directly induced Pysin expression, while CO-IP data determined

that circCDC42 overexpression also increased the assembly of Pyrin inflammasome (see Figure 6a and 6e, line 288-290 and line 296-300). These results suggested that CDC42-165aa played a dual role on Pyrin function.

In vivo experiments, we found that circCDC42 silencing inhibited the expression of Pyrin, which may be related to the reduction of inflammation in mice. In vivo, we just verified its phenotypic function due to experimental limitations. The use and development of appropriate animal models may help better define the physiological importance and contribution of CDC42-165aa, in regulating PYRIN inflammasome activity. We have thus added the statement of limitations in the Discussion section.

10) Regarding Figures 5f and 5g, demonstrating the binding of DOCK8 and CDC42-165aa is insufficient to confirm that DOCK8 is crucial for inflammasome activation and cell death. The authors should consider providing DOCK8 knockout or knockdown data to support their claims.

Response: Thank you for your suggestion. In this revised manuscript, we measured cell viability and the secretion of supernatant IL-1 β in DOCK8 knockdown MH-S cells, during KP infection. Data showed that knocking down DOCK8 significantly increased cell pyroptosis and IL-1 β release, demonstrating that DOCK8 is crucial for KP-induced inflammasome activation and cell pyroptosis (Figures S5c-S5e).

11) The manuscript should address the potential ambivalent role of CDC42 in regulating pyrin inflammasome activity, as reported in a 2022 study. The authors should discuss whether CDC42-165aa can induce Pyrin inflammasome activation solely and consider the effects of interfering with CDC42 and overexpressing CDC42-165aa.

Response: Previous paper reported that loss of CDC42 impairs inflammasome assembly, and confirmed that Cdc42 GTPase enzyme activity is upstream of Pyrin phosphorylation, and is independent of the downstream role of CDC42 in promoting inflammasome function¹.

In the revised manuscript, we found that absence of CDC42 directly reduced the

expression of Pyrin inflammasome, and exogenous complementation of CDC42-165aa just partially restored Pyrin inflammasome expression (Figures S6g). This data revealed that the regulation of CDC42-165aa in Pyrin inflammasome-mediated pyroptosis is achieved through parental gene CDC42. Compared to the previous study¹, we proposed that CDC42-165aa is involved in Pyrin inflammasome activation, while CDC42-165aa alone is not sufficient for robust Pyrin inflammasome activation. The parent gene CDC42 appears to be both necessary and sufficient for Pyrin inflammasome activation, and CDC42-165aa acts as a molecular switch in this process. Further studies are needed to unravel the complex involvement of CDC42 and CDC42-165aa in the activation of inflammasomes. We will pursue this as a part of our follow-up work further dissecting the mechanisms.

Minor Comments:

1) The resolution of images in Figures 3d and 4e is inadequate for observing dying cells undergoing pyroptosis. Improved image quality is necessary.

Response: We have improved the image quality of pyroptotic cell in Figures 3d and Figures 4e (Figures 3d and 4c).

2) The manuscript should undergo thorough proofreading to correct grammatical and syntax errors, which can impact its readability and professionalism.

Response: In the revised manuscript, we asked native speakers to correct the language problems.

In conclusion, there are significant limitations, inconsistencies, and areas of concern that need to be addressed. The authors should provide additional data, clarification, and context to strengthen the manuscript's scientific rigor and impact.

Response: We thank the reviewer for this detailed summary that clearly lays out the revisions required to improve the strengths of the manuscript. We hope that with the revisions added, we have amended the major concerns.

REFERENCES

1. Spel, L. *et al.* CDC42 regulates PYRIN inflammasome assembly. *Cell Rep* **41**, 111636 (2022).
2. Ai, Y.L. *et al.* Mannose antagonizes GSDME-mediated pyroptosis through AMPK activated by metabolite GlcNAc-6P. *Cell research* **33**, 904-922 (2023).
3. Wei, Z. *et al.* Baicalin inhibits influenza A (H1N1)-induced pyroptosis of lung alveolar epithelial cells via caspase-3/GSDME pathway. *Journal of medical virology* **95**, e28790 (2023).
4. Wree, A. *et al.* NLRP3 inflammasome activation results in hepatocyte pyroptosis, liver inflammation, and fibrosis in mice. *Hepatology (Baltimore, Md.)* **59**, 898-910 (2014).
5. Busch, C.J., Favret, J., Geirsdóttir, L., Molawi, K. & Sieweke, M.H. Isolation and Long-term Cultivation of Mouse Alveolar Macrophages. *Bio-protocol* **9** (2019).
6. Subramanian, S. *et al.* Long-term culture-expanded alveolar macrophages restore their full epigenetic identity after transfer in vivo. *Nature immunology* **23**, 458-468 (2022).
7. Hu, L. *et al.* Chemotherapy-induced pyroptosis is mediated by BAK/BAX-caspase-3-GSDME pathway and inhibited by 2-bromopalmitate. *Cell Death Dis* **11**, 281 (2020).
8. Cano, V. *et al.* Klebsiella pneumoniae survives within macrophages by avoiding delivery to lysosomes. *Cell Microbiol* **17**, 1537-1560 (2015).
9. Wanford, J.J. *et al.* Interaction of Klebsiella pneumoniae with tissue macrophages in a mouse infection model and ex-vivo pig organ perfusions: an exploratory investigation. *The Lancet. Microbe* **2**, e695-e703 (2021).
10. Dong, G. *et al.* Anthocyanin Extract from Purple Sweet Potato Exacerbate Mitophagy to Ameliorate Pyroptosis in Klebsiella pneumoniae Infection. *International journal of molecular sciences* **22** (2021).
11. Hua, K.F. *et al.* Capsular Polysaccharide Is Involved in NLRP3 Inflammasome Activation by Klebsiella pneumoniae Serotype K1. *Infection and immunity* **83**, 3396-3409 (2015).
12. Li, R. *et al.* Lyn prevents aberrant inflammatory responses to Pseudomonas infection in mammalian systems by repressing a SHIP-1-associated signaling cluster. *Signal transduction and targeted therapy* **1**, 16032 (2016).
13. Li, R. *et al.* MEG3-4 is a miRNA decoy that regulates IL-1 β abundance to initiate and then limit inflammation to prevent sepsis during lung infection. *Science signaling* **11** (2018).
14. Hua, K.F. *et al.* Capsular Polysaccharide Is Involved in NLRP3 Inflammasome Activation by Klebsiella pneumoniae Serotype K1. *Infection and immunity* **83**, 3396-3409 (2015).

REVIEWERS' COMMENTS

Reviewer #1 (Remarks to the Author):

The authors have adequately responded to the concerns raised.

Reviewer #2 (Remarks to the Author):

In response to my concerns and comments, Xu and associates have performed additional experiments and have considerably revised their manuscript (NCOMMS-23-40201A) on the role of circularCDC42 mRNA (encoding for a truncated form of CDC42 with immunomodulatory functions) in macrophages during *Klebsiella pneumoniae* infection.

My concerns of the quality of many Western blots and on the proof that circCDC42 regulates macrophage pyroptosis have been addressed appropriately. Moreover, further details have been provided on the methods and replicates. Based on this, I agree with the conclusions of this study and recommend acceptance of the manuscript.

Reviewer #3 confirmed privately to the editor that the authors had addressed most of the comments. No further remarks.

Reviewer's comments

Reviewer #1:

The authors have adequately responded to the concerns raised.

Response: We would like to thank you very much for your recognition of our work and valuable comments.

Reviewer #2:

*In response to my concerns and comments, Xu and associates have performed additional experiments and have considerably revised their manuscript (NCOMMS-23-40201A) on the role of circularCDC42 mRNA (encoding for a truncated form of CDC42 with immunomodulatory functions) in macrophages during *Klebsiella pneumoniae* infection.*

My concerns of the quality of many Western blots and on the proof that circCDC42 regulates macrophage pyroptosis have been addressed appropriately. Moreover, further details have been provided on the methods and replicates. Based on this, I agree with the conclusions of this study and recommend acceptance of the manuscript.

Response: We feel that the results obtained following your suggestions have strengthened the manuscript and we would like to thank you for your time and efforts in considering this work. We would also like to thank you for acknowledging our work and the effort we have put into the revision.

Reviewer #3

confirmed privately to the editor that the authors had addressed most of the comments.

No further remarks.

Response: Thank you very much for your positive comments and constructive suggestions, which are really helpful for the improvement of this manuscript.